# Mechanomimetic 3D Scaffolds as a Humanized In Vitro Model for Ovarian Cancer

**DOI:** 10.3390/cells11050824

**Published:** 2022-02-26

**Authors:** Francesca Paradiso, Stefania Lenna, S. Andrea Gazze, Jezabel Garcia Parra, Kate Murphy, Lavinia Margarit, Deyarina Gonzalez, Lewis Francis, Francesca Taraballi

**Affiliations:** 1Center for Musculoskeletal Regeneration, Houston Methodist Academic Institute, Houston Methodist Research Institute, 6670 Bertner Ave., Houston, TX 77030, USA; fparadiso@houstonmethodist.org (F.P.); slenna@houstonmethodist.org (S.L.); 2Orthopedics and Sports Medicine, Houston Methodist Hospital, 6445 Main St., Houston, TX 77030, USA; 3Reproductive Biology and Gynaecological Oncology Group, Swansea University Medical School, Singleton Park, Swansea SA2 8PP, UK; s.a.gazze@swansea.ac.uk (S.A.G.); j.garciaparra@swansea.ac.uk (J.G.P.); laviniamarg@doctors.org.uk (L.M.); d.gonzalez@swansea.ac.uk (D.G.); l.francis@swansea.ac.uk (L.F.); 4Department of Pathology, Singleton Hospital, Swansea Bay University Health Board, Swansea SA2 8QA, UK; kate.murphy@wales.nhs.uk

**Keywords:** 3D model, extracellular matrix, collagen, matrix stiffness, cancer, microenvironment, nanoparticles, doxorubicin

## Abstract

The mechanical homeostasis of tissues can be altered in response to trauma or disease, such as cancer, resulting in altered mechanotransduction pathways that have been shown to impact tumor development, progression, and the efficacy of therapeutic approaches. Specifically, ovarian cancer progression is parallel to an increase in tissue stiffness and fibrosis. With in vivo models proving difficult to study, tying tissue mechanics to altered cellular and molecular properties necessitate advanced, tunable, in vitro 3D models able to mimic normal and tumor mechanic features. First, we characterized normal human ovary and high-grade serous (HGSC) ovarian cancer tissue stiffness to precisely mimic their mechanical features on collagen I-based sponge scaffolds, soft (NS) and stiff (MS), respectively. We utilized three ovarian cancer cell lines (OVCAR-3, Caov-3, and SKOV3) to evaluate changes in viability, morphology, proliferation, and sensitivity to doxorubicin and liposomal doxorubicin treatment in response to a mechanically different microenvironment. High substrate stiffness promoted the proliferation of Caov-3 and SKOV3 cells without changing their morphology, and upregulated mechanosensors YAP/TAZ only in SKOV3 cells. After 7 days in culture, both OVCAR3 and SKOV3 decreased the MS scaffold storage modulus (stiffness), suggesting a link between cell proliferation and the softening of the matrix. Finally, high matrix stiffness resulted in higher OVCAR-3 and SKOV3 cell cytotoxicity in response to doxorubicin. This study demonstrates the promise of biomimetic porous scaffolds for effective inclusion of mechanical parameters in 3D cancer modeling. Furthermore, this work establishes the use of porous scaffolds for studying ovarian cancer cells response to mechanical changes in the microenvironment and as a meaningful platform from which to investigate chemoresistance and drug response.

## 1. Introduction

The microenvironment surrounding cancer cells has been shown to contribute to tumor progression at both biological and physical levels [1,2]. Indeed, tumor biophysics encompasses physical forces, such as compression, tension, hydrostatic pressure, and shear, all thought to be crucial factors in tumor microenvironment (TME) driven cancer cell sensing [3,4,5]. Alterations in extracellular matrix (ECM) composition and organization, as well as crosstalk with the surrounding physical and biochemical niche microenvironment, has been shown to drive cancer cell responses, such as proliferation, cytoskeleton distribution, migration, gene expression, and signal transduction [6,7,8,9,10,11,12]. Furthermore, mounting evidence suggest that not only the composition of the ECM, but also its stiffness, can significantly affect cancer cell responses to treatment and chemoresistance [13,14,15].

The field of cancer biomechanics aims to better understand not only how cancer cell behavior is affected by mechanical changes in the microenvironment, but also how tissue mechanical features can be exploited to detect specific disease stages, while enabling the discovery of potential new diagnostic tools and/or therapies [16]. The role of mechanical constrains in tumorigenesis has been well-studied in breast cancer, for example, where the stiffening and remodeling of the ECM accompany the promotion of breast carcinoma cell proliferation, and local tumor cell invasion and progression [17,18]. Indeed, increased density and reorganization of collagen fibrils around malignant breast tumors appears to facilitate local tumor cell invasion [19,20]. As a result, nonlinear optical imaging methods, such as multiphoton microscopy (MPM) and second harmonic generation (SHG) imaging, have been used to visualize local changes in collagen fibril density around invasive breast tumors [21].

Similar to breast cancer, epithelial ovarian cancer (EOC) evaluation with MPM and SHG imaging has revealed altered collagen fibril density and topology linked to increased stiffness and fibrosis, and associated with both primary and disseminated EOC [22,23,24]. Some evidence suggests that EOC tissue mechanical changes may also result from inflammation in the form of endometriosis [25,26]. The origin of ovarian carcinoma is still under debate, postulated to derive from any of three potential sites: the surfaces of the ovary, the fallopian tube, or the mesothelium-lined peritoneal cavity [27,28]. To invade the surrounding tissue and metastasize, ovarian carcinoma cells undergo an epithelial-to-mesenchymal transition; then, carried by the peritoneal fluid (ascites), they form multicellular aggregates (metastatic unit) called spheroids [29], overcome anoikis, and attach preferentially onto the abdominal peritoneum or omentum, a principal physiologic target for EOC dissemination [30,31]. The most common form of ovarian carcinoma is high-grade serous (HGS), usually diagnosed at an advanced stage (stage III, 70% of cases [32]), and is an inherently aggressive malignancy, thus accounting for the majority of ovarian cancer deaths [33,34]. At this late disease stage, chemotherapy resistance occurs, for reasons as yet unknown [35].

Carboplatin with paclitaxel represents the standard first-line chemotherapy regimen for ovarian cancer patients; however, only 40–60% of patients will achieve complete remission, with a high risk of neurotoxicity, which can persist for more than a year after the treatment [36,37]. Consequently, other more efficacious or tolerable options were evaluated, i.e., pegylated liposomal doxorubicin (PLD). This is an anthracycline encapsulated within a sterically stabilized liposome that increases the agent’s circulating half-life in the body and limits its toxicity profile, significantly lowering cardiac toxicity and myelosuppression compared to conventional doxorubicin [38]. It is now a widely used agent for the treatment of patients with recurrent or refractory ovarian cancer, although there are not many implications for its use as a monotherapy regimen [39,40,41].

Increased matrix stiffness is closely linked to tumor progression [42,43]; however, the malignancy of metastatic ovarian cancer has been shown to increase on soft matrices. Indeed, ovarian cancer cells on soft matrices are more proliferative and more resistant to standard chemotherapeutic drugs [31]. Since ovarian cancer mechanics and biophysics studies have resulted in contradictory findings, the exploration of mechanotransduction within ovarian cancer remains understudied [25]. Furthermore, many of the mechanical studies in EOC have been performed on spheroids and 2D polyacrylamide gels [44,45]; alternatively, the inclusion of mechanical constraints while designing an in vitro model to fully mimic native tumor tissue biology, requires the use of 3D culture platforms [46]. Rapid advances in 3D cell culture systems now allow for the recapitulation of cell differentiation and tissue organization, opening new possibilities for studying the underlying biochemical and biomechanical signals between cancer cells and the TME [47,48].

Tissue engineering (TE) describes the process of fabricating functional 3D tissues using a combination of scaffolds and/or devices with cells to facilitate essential cellular functions, such as growth, differentiation, migration, and organization [49]. In the field of regenerative medicine, these 3D devices aim to replace or “regenerate” human cells, tissues, or organs to restore or establish normal function [50]. As established by decades of research on tissue engineering (TE) and manufacturing, in order to create an effective 3D construct, three crucial components (called the TE triad) are needed: a relevant selection of cells, a biomaterial scaffold that provides the structural support for cell attachment and guides tissue development [51,52], and chemicals and biophysical signals that crosstalk to ultimately recreate tissue [53,54]. Typically, in tissue engineering, three individual groups of biomaterials are used in the fabrication of scaffolds: ceramics, synthetic polymers, and natural polymers [49]; these groups have been explored for a variety of applications, such as tissue engineering in bone [55], skin [56], cardiac tissue [57], skeletal muscle [58], and cancer models [59]. Natural biomaterials are bioactive, biodegradable, and allow host cells to produce their own extracellular matrix and remodel the scaffold [49]. However, depending on the scaffold employed, they generally possess poor mechanical properties, which limits their use in, for example, load bearing orthopedic applications.

For its high biocompatibility and bioactivity, the natural polymer collagen was selected for scaffold fabrication in this research. Collagen is the most abundant structural protein in the connective tissues, and its homology across species provides low antigenicity and high biocompatibility [60,61]; in humans, collagen represents one-third of the total protein content in the body [62]. Over the last two decades, four major scaffolding approaches for TE have evolved: pre-made porous scaffolds, decellularized extracellular matrix (ECM), cell sheets with self-secreted ECM, and cell encapsulation in a self-assembled hydrogel matrix [63]. Among these, the most common approach is the use of a pre-made porous scaffold [54], since it harbors a number of advantages: it has the most diversified range of biomaterials available to use, natural or synthetic [48]; precise architectural features and microstructures can be incorporated [64]; physicochemical characteristics can be tuned to mimic the physical properties of native tissues [65]. Specifically, matrix stiffness cues can be easily tuned in porous collagen type I-based interconnected scaffold systems by varying crosslinking types or percentages [66,67,68,69] to control porosity and fiber organization, resulting in a tunable system for 3D mechanical studies [70,71]. Indeed, easy to reproduce, convenient to handle, and amenable to large-scale use, porous scaffolds now have a wide scope of applications [70,72,73,74]. However, only a few solid tumors have been tested using these approaches, i.e., breast, prostate, and glioblastoma, mainly investigated using chitosan–alginate- or chitosan–hyaluronic acid-based scaffolds [75,76,77,78,79,80,81].

Using a sponge scaffold 3D collagen-based culture system, we investigate the role of substrate stiffness in affecting EOC cell behavior and chemoresistance in vitro. Following the assessment of fresh OC tissue to define the stiffness parameters of both peritoneum/cancer and normal ovarian mechanical features, we mimic the stiff (MS, metastatic scaffold) and soft (NS, normal scaffold) tissue properties using 3D porous matrices, testing their utility and suitability for reproducing in vivo tissue mechanics while serving as platform for drug testing. Detailed mechanical tests are performed using multiple technologies, from atomic force microscopy (AFM), easily run on the cell/tissue surface [82], to rheology and tensile/compression testing [83], performed on the bulk material. Three human EOC cell lines derived from the ovary (OVCAR-3, Caov-3) or peritoneal ascites (SKOV3) are seeded and cultured in 3D, and monitored for their ability to sense, colonize, proliferate, and remodel the collagen-rich scaffold environment. Following successful culture, EOC cell matrix models are tested for their sensitivity to doxorubicin and liposomal doxorubicin [84]. Our results show how proliferation and mechanosensing response(s) to microenvironment stiffness is EOC cell line specific and, on the other hand, higher sensitivity to chemotherapy is a common effect promoted by stiff matrices across all cell lines used. The combined data presented support the adoption of sponge scaffold models for cancer research to closely study tissue mechanical cues and their effect on ovarian cancer cells, delineating microenvironment components and testing new treatment strategies in a cost-effective and timely manner.

## 2. Materials and Methods

### 2.1. Scaffold Preparation

Chemicals were purchased from Sigma-Aldrich. The scaffolds were synthesized from type I bovine collagen (Viscofan, Cáseda, Spain) and fabricated with the freeze-dry technique. Briefly, we prepared an acetic collagen slurry (200 mg per mL, which was precipitated to a pH of 5.5 with NaOH (2N). The wet slurry was crosslinked in an aqueous solution of 0.1% *w*/*v* (normal scaffold) and 1% *w*/*v* (metastatic scaffold) BDDGE at 4 °C for 24 h. Finally, the slurry was washed with Milli-Q water (EMD Millipore, Burlington, MA, USA) and casted onto a 48-well plate and freeze-dried via an optimized freezing and heating ramp (from 25 °C to −25 °C and from −25 °C to 25 °C for 50 min under vacuum conditions, *p* = 0.20 mbar) to obtain the desired pore size and porosity.

### 2.2. Atomic Force Microscopy

The atomic force microscope used in this experiment was the Bio-Catalyst AFM (Bruker, Burlington, MA, USA). A spherical cantilever (Novascan, Chicago, IL, USA) was used for the force measurement. For biopsy samples, the QNM in fluid mode was used with a borosilicate tip (5 µm in diameter). For scaffolds, a silica bead (5 µm in diameter) was mounted onto the end of the cantilever. The spring constant of the cantilever was 0.06 N. Prior to the AFM experiment, both the spring constant and sensitivity of the cantilever were calibrated under thermal tune conditions with the controlling software (Bruker, Catalyst NanoScope 8.15 SR3R1, http://nanophys.kth.se/nanophys/facilities/nfl/afm/icon/bruker-help/Content/SoftwareGuide/NanoScope815CoverPage.htm, accessed on 3 February 2022). For the AFM experiment, both biopsies and scaffolds were embedded in OCT and cryosectioned at 20 µm section thickness. The samples were pre-coated on a glass slide and kept at −80 °C. Then, samples were carefully moved into a 60 mm dish on the AFM scanning stage. A volume of 3 mL phosphate-buffered saline (PBS) or media was pre-injected into the dish after sample incubation for 5 min at room temperature. For force measurement, we kept the ramping size at 10 µm. All experiments were conducted at room temperature (22 °C). Young’s modulus was calculated from the force curves with NanoScope Analysis 1.40 (Bruker, v1.40r1, http://nanoscaleworld.bruker-axs.com/nanoscaleworld/forums/t/812.aspx, accessed on 3 February 2022), with 3 to 5 spots randomly tested per sample and recorded, and at least 50 force curves acquired from each spot. For Young’s modulus calculations, extended ramp force curves, and a linearized model (Hertzian, spherical) were used.

### 2.3. Rheology

Wet and dry scaffolds of 1 mm thickness and 8 mm diameter were analyzed using an Anton Paar/MCR 302 rheometer equipped with an aluminum 8 mm insert plate. Both empty scaffolds and cellularized scaffolds collected at days 1–7 were characterized. An amplitude sweep test (log ramp 0.001% to 10%, angular frequency of 10 Hz, 25 recorded points, T of 37 °C) was used to verify the range of linear viscoelasticity. Frequency response was measured by frequency sweep tests in the range 1000 to 0.1 rad per s (shear strain of 0.1%, 40 data points, T of 37 °C). Storage modulus and loss moduli measures were reported as 3-sample averages collected at 1 rad per s angular frequency.

### 2.4. Compression Test

NS and MS scaffolds of 0.5 cm thickness were soaked in PBS and loaded onto a UniVert Mechanical Test System. A load cell of 1 N was calibrated and used to perform a compression test with a stretch magnitude of 35%, stretch duration of 60 s, and relaxation time of 60 s. For each condition, 3 replicates were analyzed.

### 2.5. Scanning Electron Microscopy 

The morphology of the scaffold was characterized by SEM and the pore size determined by ImageJ (US National Institutes of Health). Scaffolds were coated with 7 nm of Pt/Pl (FEI Company, Hillsboro, OR, USA, Nova NanoSEM 230) for SEM examination. The pore diameter of scaffolds was measured from SEM images, and 3 images from each of 3 areas were used for each scaffold at the same magnitude. For each image, porosity analysis was performed using ‘analyze particles’ measurement in ImageJ software (National Institutes of Health and the Laboratory for Optical and Computational Instrumentation (LOCI, University of Wisconsin), version 1.41, https://imagej.nih.gov/ij/, accessed on 3 February 2022).

### 2.6. Fourier Transform Infrared Spectroscopy

The samples were analyzed in attenuated total reflection (ATR) mode at 2 cm^−1^ resolution 64 times over the range of 500–4000 cm^−1^ using a Nicolet 6700 spectrometer (Thermo Fisher Scientific, Waltham, MA, USA). The ATR/Fourier transform infrared spectroscopy (FTIR) spectra were reported after background subtraction, baseline correction, and normalization on Amide I. Graphs reported a range of 500–1800 cm^−1^ wavelength.

### 2.7. Cell Culture 

OVCAR-3, Caov-3, and SKOV3 cells were purchased from ATCC. Cultures were established in standard growth medium, as suggested on the ATCC website, composed of RPMI-1640 Medium with 0.01 mg per mL bovine insulin and fetal bovine serum to a final concentration of 20% for OVCAR-3, Dulbecco’s Modified Eagle’s Medium with fetal bovine serum to a final concentration of 10% for Caov-3, and McCoy’s 5a Medium modified with fetal bovine serum to a final concentration of 10% for SKOV3. All growth media were supplemented with 1% penicillin (100 UI per mL)–streptomycin (100 mg per mL).

Adherent cells were detached from plates using trypsin before reaching confluence (80%) and subsequently re-plated for culture maintenance. For maintenance of cultures, cells were incubated at 37 °C in a humidified atmosphere (90%) with 5% CO_2_. Medium was changed twice per week. When seeded onto scaffolds, ovarian cancer cells were harvested and re-suspended in cell culture medium. A 20 µL drop of medium containing 1 × 10^5^ cells was seeded on the center of each scaffold and kept in an incubator for 30 min. Culture medium was then added to each well.

### 2.8. Biopsy Samples

Ethical approval for this study was obtained from NHS HRA Wales6 REC (15/WA/0065) to collect tissue samples from ovarian cancer patients and non-cancer controls. Formal written consent was obtained from all patients at the time of recruitment into the study.

A total of 6 ovarian biopsies were collected for this study and processed for AFM analyses. The histological evaluation of the ovarian biopsies and the cancer diagnosis was confirmed by the pathology department as part of the patient’s routine clinical care. Three were used as normal ovary controls as they were obtained from normal contralateral ovaries of patients diagnosed with benign pathologies. The remaining 3 ovarian biopsy tissue samples represented HGSC stage IIIc (as summarized in Table 1).

### 2.9. H&E Staining of Patient Biopsy Samples

Paraffin sections were cut at 4 μm thickness. Hematoxylin and eosin staining was performed using the ST Infinity H&E Staining System (Leica Biosystems, Wetzlar, Germany) in Leica Autostainer ST5010 XL. Paraffin was melted prior to staining by heating the slides at 60 °C for 30 min, then slides were deparaffinized by performing 3 × 2 min washes in xylene, 3 × 1 min washes in 100% ethanol, and 1 × 1 min wash in 95% ethanol, before rinsing with tap water. Slides were incubated for 30 s in Hemalast, for 5 min in hematoxylin, and were rinsed for 1 min with tap water. Next, slides were incubated for 30 s in a differentiator and 1 min in bluing agent, with each step followed by a tap water rinse for 1 min then 95% ethanol for 1 min. Slides were stained with eosin for 30 s, dehydrated in 95% ethanol for 1 min, 4 min in 100% ethanol, and 2 × 1 min in 100% ethanol, and cleared for 3 × 2 min in xylene. Every step after the initial heating of the slides was conducted at room temperature.

### 2.10. Microscopy 

#### 2.10.1. Live–Death Imaging

After 7 days of culture, scaffolds with cells were incubated with 2 µL of 50 µM calcein AM working solution and 4 µL of ethidium homodimer-1 stock and incubated for 20 min at 37 °C protected from the light. After several washes with warm media, cells were analyzed by Keyence BZX800 using a 4× objective and a final stich process to show the entire scaffold surface.

#### 2.10.2. F-Actin Imaging

After 7 days of culture, scaffolds were collected and washed with 1% PBS. After fixation with 4% paraformaldehyde for 10 min at room temperature, cellularized scaffolds were washed twice with PBS +0.1% Tween and permeabilized using Triton X-100 0.1% in PBS for 10 min at room temperature. Incubation with phalloidin-555 (Aa34055, Thermo Fisher Scientific, 1:100) and Draq-5 5 Mm (62251, Thermo Fisher Scientific, 1:20) was performed for 2 h at room temperature protected from light. After washing with PBS +0.1% Tween twice, imaging was conducted with the Nikon A1 confocal imaging system, using a 20× objective. Volume recording was performed on z-stacks of 200 μm and step size of 10 μm.

### 2.11. Flow Cytometry

#### Cell Survival and Death Quantification

After 7 days of TGFβ1 treatment, scaffolds were incubated for 10 min in trypsin under shaking conditions; subsequently, scaffolds were removed, and cell pellets were collected after centrifugation and washed with PBS. Cells were incubated with 2 µL of 50 µM calcein AM working solution and 4 µL of ethidium homodimer-1 stock, and incubated for 20 min at 37 °C protected from the light. After several washes with warm media, cells were analyzed by flow cytometry using BD FACS Fortessa.

### 2.12. Reverse Transcription Quantitative PCR

Reverse transcription PCR was performed on cells grown in 2D culture and on 3D scaffolds after 7 days. Scaffolds with cells were washed in PBS and incubated with 1 mL TRIzol RT for 10 min under shaking conditions. After removing the scaffolds, 200 µL of chloroform was added and samples were inverted for 15 min, incubated on ice for 2 min, and centrifuged at 12,000× *g* for 15 min at 4 °C. The aqueous phase was transferred to a 1.5 mL tube and 500 µL of isopropyl alcohol added, before incubating for 10 min at 4 °C and centrifuging 12,000× *g* for 10 min at 4 °C. After washing the pellet twice with 1 mL 70% ethanol, it was aspirated and allowed to dry before resuspending in 20 µL of water. Total RNA (500 ng) was reverse transcribed into cDNA using the Bio-Rad iScript™ cDNA Synthesis Kit. Quantitative PCR was performed using the TaqMan™ Fast Advanced Master Mix on a StepOnePlus Real-time PCR System (Applied Biosystems, Waltham, MA, USA). Expression of YAP1 (Hs00902712) and WWTR1 (reported as TAZ, Hs00210007_m1) was detected using TaqMan^®^ Gene Expression Assays. 18S ribosomal RNA was used as an internal reference for normalization. Analysis was performed using the relative ΔΔCT method.

### 2.13. MTT Assay Protocol for Cell Viability and Proliferation

To perform this test, SKOV3 and Caov-3 cells were seeded at a concentration of 8000 cells per well in 96-well plates, while 20,000 OVCAR-3 cells were seeded per well in a final volume of 100 µL per well. MTT was added to achieve a final concentration of 0.5 mg per mL MTT in normal media. After incubating 2 h at 37 °C, MTT was removed and 100 µL DMSO added before mixing contents for 30 min on an orbital shaker protected from light. Absorbance was measured at 590 nm.

### 2.14. CellTiter-Glo^®^ Luminescent Cell Viability Assay

For the proliferation assay, ovarian cancer cells were grown on scaffolds for 10 days and analysis performed at 4 time points (day 1, day 4, day 7, day 10). For cytotoxic evaluation of doxorubicin, ovarian cancer cells were grown on scaffolds for 7 days, and then treated with free doxorubicin (DOXO), doxorubicin-loaded liposomes (DOXO-LIPO), or empty liposomes for 72 h. To perform the analysis, a volume of CellTiter-Glo^®^ Reagent (Promega, Madison, WI, USA) equal to the volume of cell culture medium present in each well/scaffold was added. Contents were mixed for 10 min on an orbital shaker to induce cell lysis, protected from light. The plate was incubated at room temperature for 25 min to stabilize luminescent signal, before transferring 100 µL (or a 1:10 dilution in media for high signals) in white opaque-walled 96-well plates to measure luminescence.

### 2.15. Assembly and Physical Characterization of Liposomes

To assemble liposomes, 20 mg of total lipids including DPPC, DSPC, DOPX, and cholesterol (molar ratio 5:1:3:1) were dissolved in methanol–chloroform solution (1:3 *v*/*v*) to a final volume of 1 mL. The solvent was evaporated via a rotary evaporator (Buchi Labortechnik AG, Flawil, Switzerland) for 20 min at 45 °C to form a thin lipid film. The film was hydrated with 1 mL sterile water to assemble empty liposomes or 250 mM ammonium sulfate for liposomes to be loaded with doxorubicin. The 1 mL solution was incubated for 3 min at 45 °C followed by 3 min vortexing. Lipid suspension was forced through a polycarbonate filter (200 nm; GE Osmonics Labstore, Minnetonka, MN, USA) 10 times under nitrogen gas pressure at 45 °C (filter was replaced after 5 extrusions). Size, zeta potential, and polydispersity index (PDI) were measured using dynamic light scattering. After the nanoparticles (NPs) were fabricated, they were loaded into dialysis floaters in order to exchange the outside buffer with 0.9% NaCl overnight. Lipid formulation was then incubated at 1:1 *v*/*v* for 2 h at 45 °C with 1 mL of 2 mg per mL doxorubicin hydrochloride (Sigma-Aldrich, St. Louis, MO, USA, D1515) dissolved in DDW. Only DOXO-LIPO NPs were loaded again into dialysis floaters in order to exchange the outside buffer with 0.9 NaCl overnight. NanoSight NS300 (Malvern, Worcestershire, UK) for both empty and loaded NPs was performed as the final step to measure lipid NPs concentration.

### 2.16. Evaluation of Doxorubicin Encapsulation Efficiency and Release

Doxorubicin encapsulation and drug release analysis was performed using a Tecan Microplate Reader. For the doxorubicin release experiment, DOXO-LIPO NPs were incubated with PBS +10% FBS (50:50) at 37 °C under shaking conditions and samples were collected and analyzed after 0.5, 1, 1.5, 2, 4, 5, 8, 24, 48 h. For DOXO encapsulation, DOXO-LIPO NPs were diluted in water to 1:200 and mixed (1:1 *v*/*v*) with 0.2% (*v*/*v*) triton x-100 (overall doxorubicin) or water (released doxorubicin) in a black 96-well plate for 5 min at room temperature under shaking conditions. Doxorubicin fluorescence was read at excitation 480 nm/emission 590 nm and cyclophosphamide (Cy) fluorescence at 5.5 excitation 650 nm/emission 700 nm.

### 2.17. Statistical Analysis

All data were obtained from at least 3 independent experiments and expressed as mean ± standard deviation, with *n* indicating the number of replicates. The two-tailed Student’s t test with Welch’s correction or an ANOVA test was used to determine differences between groups. Results were considered to be statistically significant at *p*-value < 0.05. The statistical analysis was processed with GraphPad Prism 6 Software (GraphPad, San Diego, CA, USA).

## 3. Results

### 3.1. Scaffold Characterization

Patient biopsy mechanics, derived from both high-grade serous carcinoma (HGSC) stage III and normal ovary tissues, were analyzed at the nanoscale using AFM. A 5.5-fold increase in stiffness was observed between HGSC IIIa (0.11 ± 0.034 MPa) and normal ovary (0.02 ± 0.016 MPa) tissues (*p* < 0.05; Figure 1A, Appendix A). The AFM data from native tissues were successfully mimicked in a 3D collagen type I-based scaffold model. Using 1% and 0.1% *w/v* BDDGE crosslinkers achieved a higher stiffness for the metastatic scaffolds (MS, 0.144 ± 0.010 MPa) compared with the normal scaffolds (NS, 0.015 ± 0.0003 MPa), respectively (*p* < 0.05; Figure 1B). H&E staining of patient biopsy samples was performed (Appendix A), reporting a high presence of ECM/fibrotic tissue in the HGSC IIIc-derived samples (Appendix A).

In addition to AFM, shear rheometry was performed on the fabricated scaffolds to gain knowledge on the resulting bulk tissue mechanics characterizing our model [85]. The scaffold elastic component, called the storage modulus (G’), at 1 Hz was significantly increased in the MS (0.011 ± 0.0006 MPa) compared to the NS (0.0036 ± 0.00046 MPa) scaffolds, showing a 3-fold increase in elastic properties; alternatively, the viscous component, called the loss modulus (G’’), was comparable between the two scaffold types (MS: 0.0011 ± 0.0001 MPa; NS: 0.0010 ± 0.0004 MPa) (Figure 1C,D). Subsequent compressive tests were carried out to evaluate the compressive strength and stiffness of the scaffolds. The results, summarized in Figure 1E, showed that higher force was required to compress MS (0.54 ± 0.028 N) compared to NS (0.16 ± 0.025 N) scaffolds.

The porous structure of MS and NS scaffolds after freeze-drying was determined by SEM imaging (Figure 2A). At lower magnification, the sample structures were composed of interconnected pores with boundaries defined by sheet-like structures of fibrillar collagen. At higher magnification, the typical fibrous substructure of collagen sponges can be appreciated. Porosity measurements showed that MS and NS scaffolds exhibited a comparable average pore size of approximately 2500 µm^2^, corresponding to an average diameter of 56.4 µm, with a comparable percentage area of 30 µm^2^ covered by pore structures, and a pore circularity of approximately 0.37 (Figure 2B).

FTIR was used to characterize scaffold composition after crosslinking. FTIR spectra, reported in Figure 2C, showed the characteristic collagen vibration peaks, such as Amide I (1700–1600 cm^−1^) and Amide II (1600–1500 cm^−1^), related to the stretching vibration of C=O bonds, and to C–N stretching and N–H bending vibrations, respectively, for both scaffold types. The samples contained C=O, C–N, and N–H bonds. The Amide III region (approximately 1200–1300 cm^−1^) is related to C–N and C–C stretching, N–H bonds, and CH2 wagging from the glycine backbone and proline side chain.

### 3.2. Viability, Morphology, and Proliferation of Ovarian Cancer Cells on MS and NS Scaffolds

We employed three cell lines derived from the ovary (OVCAR-3, Caov-3) or peritoneal ascites (SKOV3) of EOC [86] to test their phenotypical, behavioral, and transcriptional differences in response to microenvironment stiffness. First, cell death was monitored after 7 days of culture on NS and MS scaffolds, using calcein/ethidium bromide staining, assessed by epifluorescence microscopy, and quantified by flow cytometry (Figure 3). All three cell lines were able to attach to, and colonize, the scaffolds, as reported by the 3D maximum intensity projection based on three layers collected per scaffold (Figure 3A,C,E). When analyzed quantitatively for viability, calcein-positive OVCAR-3, Caov-3, and SKOV3 percentages were, respectively, 84.29% ± 6.5% (Figure 3A), 88.75% ± 10.4% (Figure 3C), and 80.83% ± 5.8% (Figure 3E) on MS and 91.39% ± 7.04%, 93.01% ± 5.3%, and 87.95% ± 4.3% on NS scaffolds, with no statistical differences in cell viability between the two scaffold types. No morphological changes were detected after 7 days of culture in the cancer cells lines on the scaffolds, as shown by F-actin immunofluorescence staining. The OVCAR-3 cells maintained their cuboidal shape (Figure 3B), Caov-3 cells their spindle-shaped morphology (Figure 3D), and SKOV3 cells their elongated spindle-shaped morphology (Figure 3F). 

Cancer cell proliferation rate was assessed by CellTiter-Glo^®^. The OVCAR-3 proliferation rates showed no differences between MS and NS scaffolds until day 10 of culturing, at which point, OVCAR-3 proliferation was observed to be higher on NS scaffolds when compared to their MS counterparts (8.6 ± 0.19 and 7.7 ± 0.5, respectively) (Figure 4A). The Caov-3 cell line showed a 1.3-, 1.2-, and 1.2-fold increase in proliferation at days 4, 7, and 10, respectively, when cultured on the MS compared to the NS scaffolds (Figure 4B). The SKOV3 cell line showed higher proliferation rates on MS scaffolds from day 7, exhibiting a 1.08-fold increase in proliferation at day 7 and 1.05-fold increase at day 10 on MS compared to NS scaffolds (Figure 4C).

### 3.3. Mechanosensing by Ovarian Cancer Cells and the Impact of the Microenvironment Mechanics

To understand how ovarian cancer cells culturing on 3D scaffolds influence overall scaffold tissue mechanics, we evaluated the bulk mechanics, using rheometry, at day 1 and day 7 of culture (Figure 5A–G). After 1 day of culturing, MS and NS scaffolds still harbored significant differences in their storage moduli (*p* < 0.01 and *p* < 0.05); for OVCAR-3-cultured scaffolds, these were 0.093 ± 0.041 MPa for MS and 0.012 ± 0.002 MPa for NS; for Caov-3, these were 0.032 ± 0.005 MPa for MS and 0.012 ± 0.004 MPa for NS; and for SKOV3, these were 0.039 ± 0.011 MPa for MS and 0.012 ± 0.0004 MPa for NS. A decrease in MS storage moduli was observed at day 7 for the OVCAR-3 culture (MS 0.030 ± 0.006 MPa, NS 0.005 ± 0.004 MPa) and SKOV3 culture (MS 0.021 ± 0.002 MPa, NS 0.007± 0.001 MPa) (Figure 5A,C,E); in contrast, no differences in storage moduli at day 7 were observed for Caov-3 (MS 0.029 ± 0.007 MPa, NS 0.012 ± 0.0004 MPa) (*p* < 0.01). Regarding SKOV3 no significant differences were recorded for loss moduli at day 7. At day 1, loss moduli for OVCAR-3 were MS 0.015 ± 0.010 MPa, NS 0.002 ± 0.0001 MPa; for Caov-3 were MS 0.004 ± 0.001 MPa, NS 0.002 ± 0.001 MPa; and for SKOV3 were MS 0.005 ± 0.002 MPa, NS 0.002 ± 0.0001 MPa (all *p* < 0.05). These values at day 7 were: OVCAR-3 MS 0.004 ± 0.001 MPa, NS 0.001 ± 0.001 MPa; Caov-3 MS 0.004 ± 0.001 MPa, NS 0.002 ± 0.0001 MPa (*p* < 0.05); SKOV3 MS 0.003 ± 0.0001 MPa, NS 0.001 ± 0.0001 MPa (Figure 5B,D,F).

Mechanical changes in the microenvironment can strongly affect cells at the transcriptomic level through a process called mechanosensing [47]. We evaluated the expression of two master mechanosensors and transcriptional activators, *Yap1* and *Taz*, which are essential for triggering cancer initiation and growth of most solid tumors [87]. Among the ovarian cancer cell lines tested, only SKOV3 reported a 2.2-fold increase in Yap1 expression and a 2.7-fold increase in Taz expression on MS compared to NS scaffolds, supporting mechanosensing-related pathway activation promoted by increased substrate rigidity (Figure 5G,H,I). This suggests a differential and specific oncogenic role of YAP [88], and its potential use as predictive factor [89], for rapidly proliferating and highly metastatic SKOV3 cell lines. Interestingly, a large body of evidence indicates that YAP/TAZ activation and overexpression is implicated in resistance to targeted therapies, chemotherapy (such as DNA damaging agents), radiation, and immunotherapies [90].

### 3.4. 3D In Vitro Cytotoxic Effect of Free Doxorubicin and Doxorubicin-Loaded Liposomes

To evaluate the link between stiffness of the substrate, mechanosensing, and resistance to chemotherapy, we employed our 3D model platforms to test ovarian cancer cell sensitivity to mainline chemotherapeutic treatments, doxorubicin (DOXO) and doxorubicin-loaded liposome (DOXO-LIPO). Similar formulations of doxorubicin have been reported to be effective, with tolerable side-effects in either combination therapy with carboplatin or in monotherapy for recurrent or platinum-resistant ovarian cancer [91].

The physicochemical features of conventional liposomes [92,93] were retained when loaded with DOXO. DOXO encapsulation did not significantly affect the size and polydispersity of DOXO-LIPO. Indeed, compared to empty liposomes (hydrodynamic size = 184 ± 2.0 nm; PDI = 0.15 ± 0.151), DOXO encapsulation did not alter nanovesicle diameter (hydrodynamic size = 183.5 ± 2.8 nm), and only slightly decreased size distribution (PDI = 0.08 ± 0.015) to values that are still below 0.2, thus indicating a high size homogeneity. Surface charge was similar between empty liposomes (Z potential = −8.4 ± 0.6) and DOXO-LIPO (Z potential = −8.42 ± 0.94) (Appendix A).

The DOXO loading efficiency into liposomes was 93.9% ± 3.63% (Appendix A), with a release kinetic of DOXO from liposomes of 16.5% after 8 h, and a 56% DOXO release after 72 h (Appendix A). The in vitro cytotoxicity of free DOXO, DOXO-LIPO, and empty liposomes was tested against OVCAR-3, Caov-3, and SKOV3 cell lines grown in 2D culture, on NS and MS scaffolds. In 2D culture, the MTT assay was used to evaluate cell viability and proliferation, and in 3D culture, growth inhibition was assessed using CellTiter-Glo^®^. After normalization against untreated cells, we observed a similar effect of free DOXO and DOXO-LIPO in reducing all ovarian cancer cell viability in 2D, while empty liposomes did not affect cell viability after either 48 or 72 h of treatment. Interestingly, OVCAR-3 cells in 2D were the most resistant to treatment after 48 and 72 h in the lower dose range, showing reductions in cell viability at 72 h of 64, 43, and 35.8% for DOXO, and 75, 46, and 42.6% for DOXO-LIPO, at concentrations of 0.0625, 0.125, and 0.25 µM, respectively (Appendix A). Meanwhile, at the same concentrations, Caov-3 cells showed reductions in cell viability of 30, 21, and 11% for DOXO, and 45, 35, and 21% for DOXO-LIPO (Appendix A), and SKOV3 cells showed reductions of 21, 10, and 8% for DOXO, and 30, 10, and 8% for DOXO-LIPO (Appendix A).

On the 3D in vitro scaffolds, cells were initially treated with a dose ranging from 0.25 to 5 μg per mL DOXO and DOXO-LIPO (data not shown), and we chose to focus our analysis on a low and high dose close to the IC_50_ value (0.25 and 1 μg per mL) (Appendix A). Overall, both 0.25 and 1 μg per mL of DOXO and DOXO-LIPO had a higher cytotoxic effect on 2D cultures rather than 3D MS and NS scaffolds. DOXO-LIPO and DOXO showed comparable effects among all cell lines tested, with the only exception of OVCAR-3 cells in 3D culture, in which free DOXO had a greater cytotoxic effect compared to DOXO-LIPO (DOXO and DOXO-LIPO, 1 μg per mL; MS 20% and 70% cell viability; NS 46% and 98% cell viability) (Figure 6A,B). Interestingly, Caov-3 was more sensitive than the other two cell lines to both DOXO and DOXO-LIPO at 1 μg per mL, exhibiting a cytotoxic effect comparable to the one obtained in 2D (DOXO 13%, 36%, and 30%; DOXO-LIPO 29%, 34%, and 37%; for 2D culture, MS and NS, respectively) (Figure 6C,D). Finally, SKOV3 showed DOXO and DOXO-LIPO cytotoxic effects at 1 μg per mL on MS (33% and 50% cell viability) and NS (57% and 63% cell viability) scaffolds, respectively (Figure 6E,F). Empty liposomes had minimal effect on the proliferation of all cancer cell lines (Appendix A).

## 4. Discussion

From a biophysical standpoint, solid tumor cancers have been shown to share physical characteristics; however, these are affected by specific mechanical cues linked to the tissue of origin [12,94,95,96]. For example, brain tumors are soft, whereas pancreatic tumors are rigid [13,97]. In such tissues, the inputs conveyed from the surrounding microenvironment are transmitted through surface receptors to the cellular compartments, and ultimately the cell nucleus, where they can influence gene expression and promote disease development and differentiation [98,99]. Tunable collagen sponge models allow cancer cell proliferation, morphology, migration, and drug response to be monitored in differential mechanical and biophysical contexts. In line with this, the use of 3D matrices to build physically meaningful platforms for drug testing is essential, since cells grown in 2D dish cultures experience a different force pattern, which could impact drug treatment outcome evaluation.

The NS and MS scaffold 3D in vitro model mimics the difference in stiffness between normal ovarian tissue and pathological HGSC stage III-derived tissue, respectively. After successfully mimicking in vivo stiffness on scaffolds MS (0.144 ± 0.010 MPa) and NS (0.015 ± 0.0003 MPa), we zoomed out from the AFM analysis, to evaluate the scaffolds’ bulk mechanic characteristics using rheology. A systematic mechanical characterization of the bulk tissue mechanics of patients’ biopsy samples, or having the diagnostic tools able to detect changes in patients’ tissue mechanics, could provide guidelines for clinical mechanopathology evaluation [85], ultimately helping doctors during diagnosis or deciding which tumors are most likely to develop chemoresistance, improving prognosis. Rheological analysis confirmed the results detected using AFM, reporting that MS scaffolds (0.011 ± 0.0006 MPa) were stiffer than NS scaffolds (0.0036 ± 0.00046 MPa), harboring a 3-fold increase in the storage modulus.

Our results are in line with previous research that analyzed tumor stiffness in vivo. For example, researchers employed supersonic shear wave elastography in a patient-derived xenograft (PDX) mouse model engrafted with HGSOC tumors isolated from patients, recording a significant increase in tumor stiffness (120 to 140 kPa) over time in mesenchymal HGSOC, while stiffness remained low (60 kPa maximum) in non-mesenchymal tumors. In high-grade serous ovarian cancers (HGSOC), representing the vast majority (75%) of total ovarian cancers, the “fibrosis” or “mesenchymal” HGSOC molecular subtype has been identified in all studies, and is systematically associated with poor patient survival. It is characterized by high stromal content composed of myofibroblasts and ECM proteins, such as collagen and fibronectin, which are major causes of tumor stiffness [100]. In the same study, in few cases, the authors observed a new tumor nodule emerging from a stiff mesenchymal tumor, interestingly, the new nodule—of small size—was softer than the established initial tumor, suggesting tumor proliferation could originate in an initial soft state. Our data are in line with observations made in other cancer types. In breast cancer, malignant tissue is typically stiffer than its normal counterpart, with studies showing that normal breast tissue is 20 times softer than its neoplastic counterpart [101]; in addition, the elastic moduli of healthy thyroid tissue (9.0–11.4 kPa) can increase by a full order of magnitude, up to 44–110 kPa, in patients with papillary adenocarcinoma [102]. Furthermore, the storage modulus of MS and NS scaffolds spans within the range of stiffness reported in the literature referring tissue/organ stiffness, ranging from 0.2–64 kPa [103,104,105].

To test the mechanoresponsive potential of ovarian cancer malignancy, we selected three adenocarcinoma ovarian cancer cell lines: OVCAR-3 and Caov-3, derived from the ovary, and SKOV3, derived from ascitic fluid from post-chemotherapy patients. OVCAR-3 and Caov-3 possess TP53 mutations and substantial copy-number changes, which are key characteristics of HGSC, whereas SKOV3, based on its genetic profile, is categorized as non-serous [106,107]. Although these cell lines can be divided into different categories by their mutation profiles, their behaviors in vitro do not necessarily segregate the same way and, most interestingly, the cell lines do not all behave as expected based on their putative identity. Indeed, cell lines derived from non-serous carcinomas (e.g., SKOV3) were shown to migrate more quickly, and were more likely to invade into Matrigel and collagen I substrates than cell lines derived from high-grade serous carcinomas (e.g., OVCAR-3) [108]. Furthermore, other researchers found that SKOV3 cells had high tumorigenicity when injected intraperitoneally, whereas OVCAR-3 cells had low tumorigenicity when inoculated in nude mice over 6 weeks [109]. For this reason, we selected relevant HGSC and non-serous tumor-derived ovarian cancer cell lines to investigate mechanosensing behavior in response to different 3D biomechanic scaffolds. All the cell lines employed in our study successfully colonized, and were viable on both MS and NS scaffolds with no changes in morphology. However, looking at the migration depth in the scaffolds recorded by immunofluorescence staining, SKOV3 and Caov-3 showed higher invasiveness compared to OVCAR-3. Indeed, previous findings suggest that changes in stiffness of the cancer cell niche, as would be encountered by disseminated or metastatic OCCs, represents the mechanism to further promote EMT [15]. A significant difference was recorded in the proliferation rate, showing higher proliferation of OVCAR-3 on NS, while both Caov-3 and SKOV3 had higher proliferation rates on MS, suggesting higher responsiveness to rigid substrates. The higher rate of proliferation of OVCAR-3 and SKOV3 compared to Caov-3 resulted in a lower storage modulus after 7 days of culturing for MS, suggesting a link between cell proliferation and softening of the scaffolds recorded with rheometry. This phenomenon could be linked to two phenomena reported in literature. First, cancer cells are physically softer than normal cells [110,111], and metastatic cancer cells are more mechanically compliant than their non-metastatic counterparts [112,113], contributing to the overall softening of the tissue. Second, previous findings suggest that a reduction in the stiffness of the cancer cell niche, as would be encountered by disseminated or metastatic OCCs, is a mechanism used to promote EMT [15], suggesting that the progressive softening of the matrix is a crucial step to promote metastasis.

Yorkie-homologues YAP (Yes-associated protein) and TAZ (transcriptional coactivator with PDZ-binding motif, also known as WWTR1) are transcriptional coactivators pervasively activated in human malignancies. Their activation in cancer cells impacts the behavior of cancer cells themselves by regulating their capacity to proliferate and adjust their metabolism to the altered cellular context, promoting the acquisition of stem-like properties, drug resistance, and migratory capacity that allow tissue invasion and metastatic dissemination. Interestingly, in the present study, YAP/TAZ are sensors of the structural and mechanical features of the cell microenvironment, including changes in mechanotransduction, inflammation, oncogenic signaling, and inhibition of the Hippo pathway [114,115]. Many correlations between high YAP and TAZ expression with poor patient outcome were reported for breast, colorectal, liver, and pancreatic cancers [87]; specifically, TAZ is thought to play an important role in breast cancer progression, with both mRNA and protein expression reported to be preferentially higher in triple-negative breast cancer than in the other subclasses [116,117,118,119]. Previous research reported that SKOV3 cells express low levels of endogenous YAP (and low activity) compared to Caov3 and OVCAR3 cells, which instead showed higher levels of both YAP expression and activity [120]. Interestingly, we observed that, while OVCAR-3 and Caov-3 downregulated YAP/TAZ expression on MS scaffolds, SKOV3 cells increased the expression of both markers on MS compared to NS scaffolds, suggesting a specific YAP/TAZ upregulation linked to a high stiffness microenvironment.

Aside from intrinsic molecular mechanisms [121], tumor chemoresistance is also affected by the biochemical and physical properties of the tumor microenvironment [122,123,124]. Specifically, the chemotherapeutic response of ovarian cancer cells in vitro is markedly affected by substrate stiffness; for example, during an evaluation of ovarian cancer tumor response to standard chemotherapeutic drugs (cisplatin and paclitaxel), antiproliferation effects were directly proportional to the stiffness of the substrate, thus, ovarian cancer cell lines grown on softer substrates with a lower elastic modulus were less sensitive to chemotherapeutic agents [15].

First-line management for ovarian cancer consists of surgery plus platinum-based combination chemotherapy, typically cisplatin or carboplatin, with the addition of a taxane, either paclitaxel or docetaxel [125]. Nevertheless, more than 70% of patients experience relapse after first-line therapy [126]. For patients that experience partially platinum-sensitive relapse (progression within 6 to 12 months after the last platinum-based chemotherapy treatment), the treatment has not yet been standardized [127,128,129,130,131].

Approved by the US Food and Drug Administration (FDA) in 1995, Doxil (doxorubicin HCl liposome injection, Tibotec Therapeutics, Division of Ortho Biotech Products, L.P.) was the first nanodrug marketed in the United States for the treatment of ovarian cancer in women for which the disease has progressed or recurred after platinum-based chemotherapy [132]. Many clinical trials showed response rates, progression-free survival, and overall survival similar to other platinum-based combinations, although with a more favorable toxicity profile and convenient dosing schedule when Doxil was tested [133,134].

In line with these results, we tested the cytotoxic effects of free DOXO and DOXO-LIPO on cis-platinum-resistant SKOV3 [135], OVCAR-3 [136], and Caov-3 [137] cell lines grown in 2D and on 3D MS and NS scaffolds. We reported a higher resistance to treatment in 3D vs. 2D culturing conditions, further stressing the importance of adopting 3D models to perform more reliable in vitro drug screening and dosing. Furthermore, among the different ovarian cancer cell lines, Caov-3 was more sensitive to both DOXO and DOXO-LIPO treatment, pointing out the necessity to include many different cell lines or primary cells when performing cancer studies evaluating potential patient–treatment responses. Finally, both OVCAR-3 and SKOV3 showed a higher resistance to treatment when grown on NS scaffolds, and a more sensitive phenotype when grown on stiffer MS scaffolds, thus confirming a previous study reporting cell growth inhibition by doxorubicin in response to ECM rigidity [138].

## 5. Conclusions

To summarize, the inclusion of physical parameters in 3D in vitro model design will help configure a microenvironment closer to native tissues, which could sustain meaningful cell culture conditions for cancer research and drug testing. Envisioning a clinical application, the use of patient-derived primary cells in combination with a 3D biomechanical scaffold, could be used to assess the likelihood of a favorable outcome to tumor treatment, and eventually suggest possible alternative patient-tailored options.

## Figures and Tables

**Figure 1 cells-11-00824-f001:**
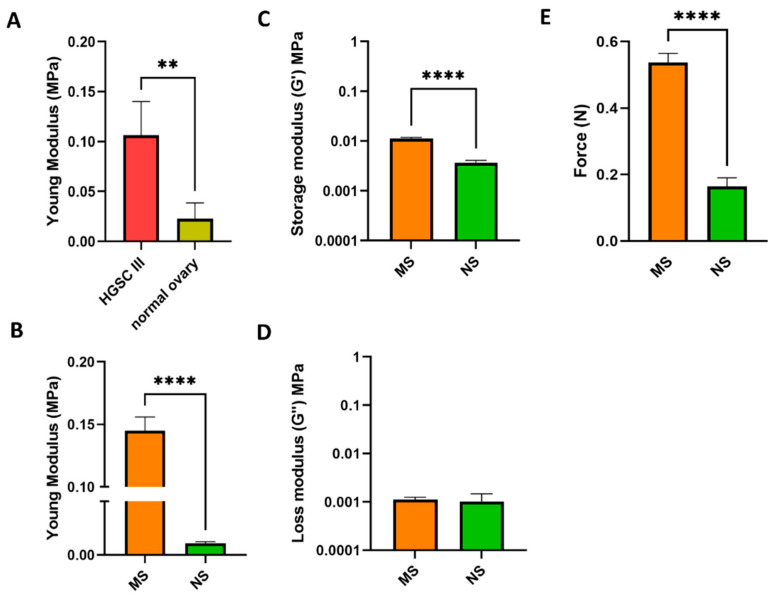
Mechanical features of normal and cancer tissues are mimicked in a 3D collagen-based in vitro system using different percentages of crosslinker 1,4-butanediol diglycidyl ether (BDDGE). (**A**) Young’s modulus (MPa) analysis of 3 HGCS III and 3 normal patient-derived biopsies by AFM. (**B**) Young’s modulus (MPa) analysis of MS and NS scaffolds by AFM. (**C**) Rheology analysis of MS and NS scaffold storage moduli (G’, MPa). (**D**) Rheology analysis of MS and NS scaffold loss moduli (G”, MPa). (**E**) Compression test analysis of MS and NS scaffolds. Data are mean + standard deviation (*n* = 3). Student’s t test with Welch’s correction, **** *p* < 0.0001, ** *p* < 0.01.

**Figure 2 cells-11-00824-f002:**
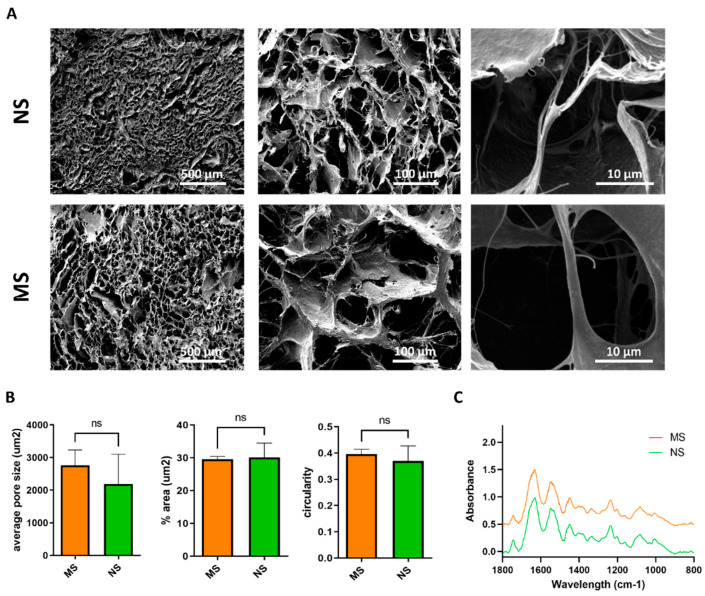
NS and MS scaffolds showed common pore sizes, pore coverage, and composition. (**A**) SEM imaging of NS and MS scaffolds at different magnifications. (**B**) SEM analysis of average pore size (µm^2^), % area covered by pore structures, and circularity. Student’s t test with Welch’s correction performed (ns: not significant). (**C**) FTIR spectra of MS and NS scaffolds. The spectra highlighted the presence of typical collagen Amide I, Amide II, and Amide III.

**Figure 3 cells-11-00824-f003:**
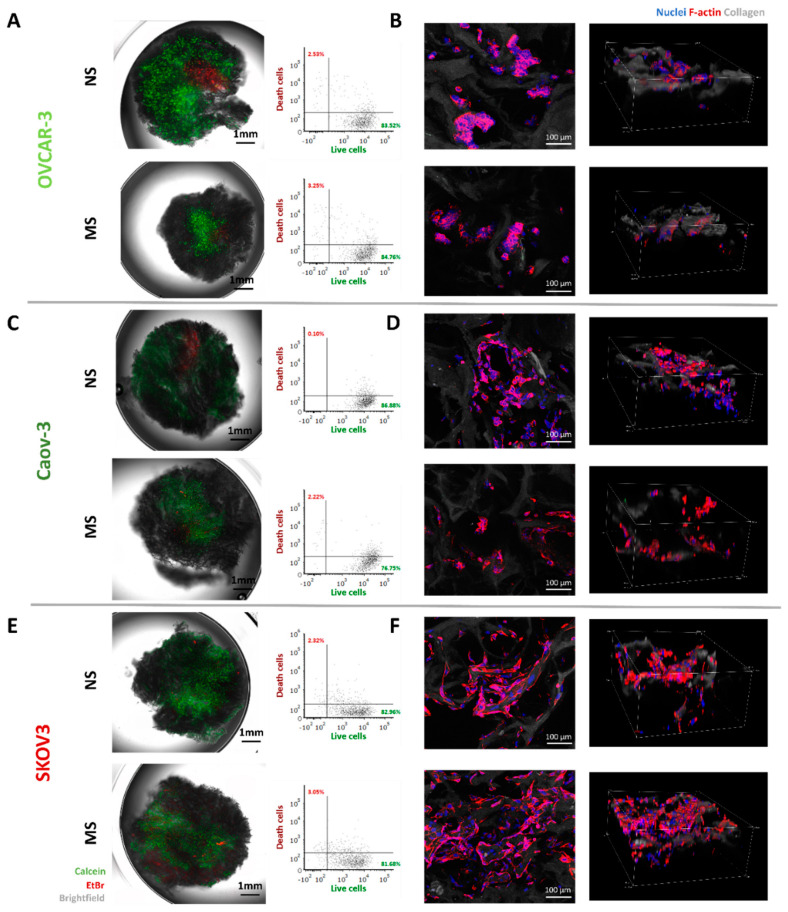
Ovarian cancer cell lines were viable and did not change their morphology when cultured on both MS and NS scaffolds. Cell death staining analyzed with fluorescence microscopy and flow cytometry of (**A**) OVCAR-3, (**C**) Caov-3, and (**E**) SKOV3 cells. Immunofluorescence staining of F-actin and DAPI in (**B)** OVCAR-3, (**D**) Caov-3, and (**F**) SKOV3 cells. Analysis and imaging was performed after 7 days of culture. In lateral projection pictures, Z-size is 200 μm and step size is 10 μm.

**Figure 4 cells-11-00824-f004:**
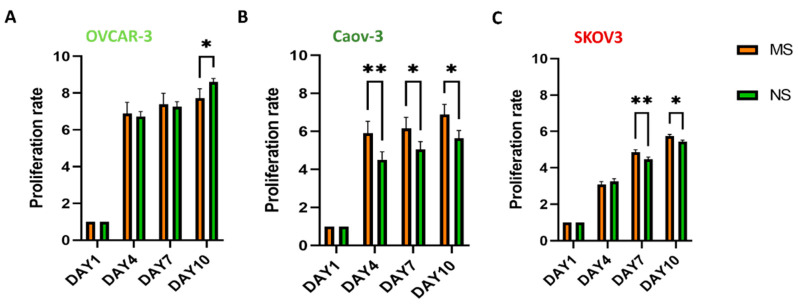
SKOV3 and Caov-3 cells proliferate more rapidly on MS scaffolds, while OVCAR-3 cells proliferate more rapidly on NS scaffolds. CellTiter-Glo^®^ analysis of ovarian cancer cell proliferation rate from day 1 to day 10 on MS and NS scaffolds for (**A**) OVCAR-3, (**B**) Caov-3, and (**C**) SKOV3 cells. Data are mean + standard deviation (*n* = 3). Statistical analysis performed with two-way ANOVA. ** *p* < 0.01, * *p* < 0.05.

**Figure 5 cells-11-00824-f005:**
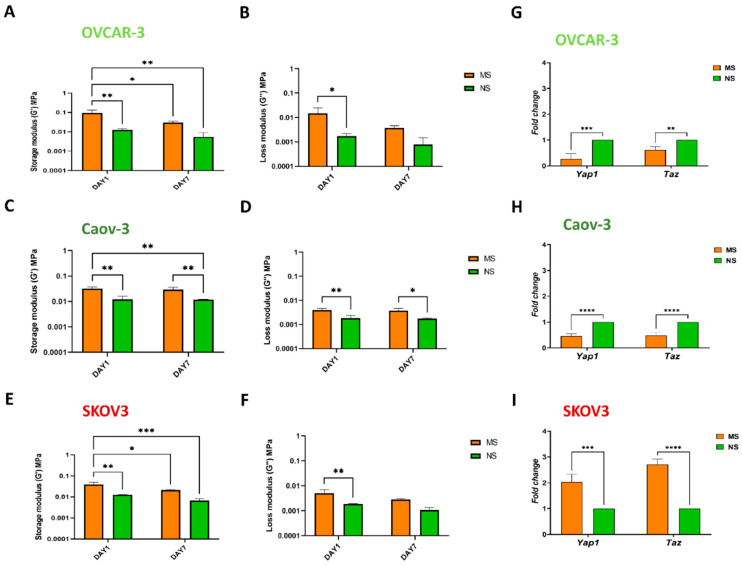
Bulk mechanical properties are slightly changed by OVCAR-3 and SKOV3, while Hippo pathway activation is specific to the SKOV3 cell line. Rheology analysis of storage modulus (G’) in MS and NS scaffolds at days 1 and 7 of OVCAR-3 (**A**), Caov-3 (**C**), and SKOV3 (**E**) culture. Rheology analysis of loss modulus (G”, right) in MS and NS scaffolds at days 1 and 7 of OVCAR-3 (**B**), Caov-3 (**D**), and SKOV3 (**F**) culture. mRNA expression of mechanosensing-related genes (*Yap* and *Taz*) in OVCAR-3 (**G**), Caov-3 (**H**), and SKOV3 (**I**)-cultured scaffolds at day 7. Data normalized to NS scaffolds. Data are mean + standard deviation (*n* = 3). Statistical analysis performed with two-way ANOVA. **** *p* < 0.0001, *** *p* < 0.001, ** *p* < 0.01, * *p* < 0.05.

**Figure 6 cells-11-00824-f006:**
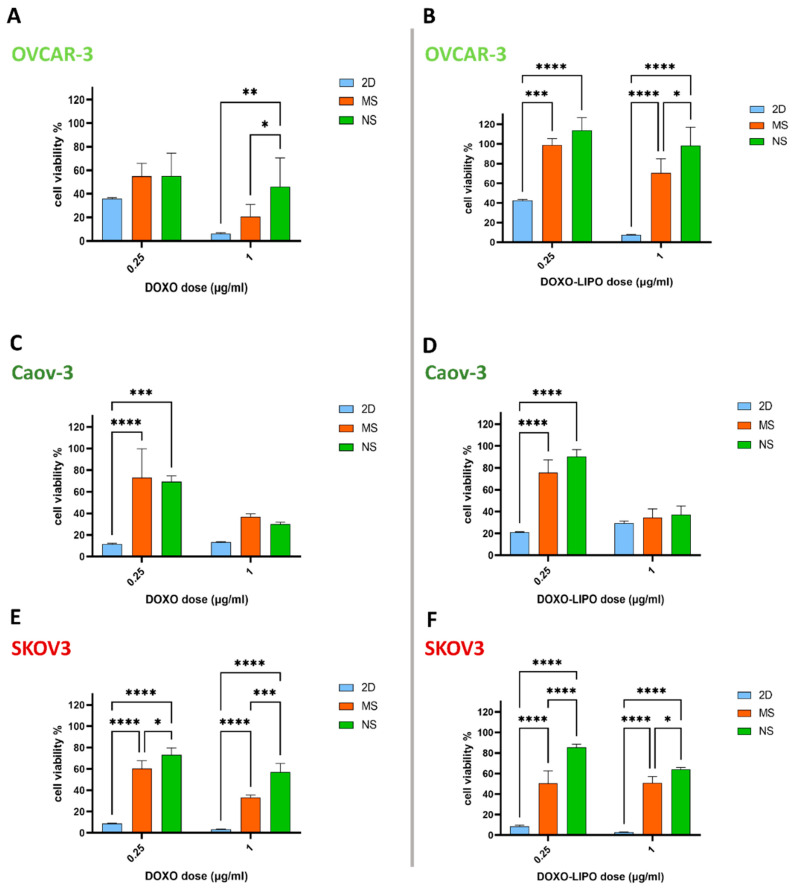
DOXO and DOXO-LIPO cytotoxic effects on ovarian cancer cell lines grown in 2D culture and on 3D MS and NS scaffolds. MTT and CellTiter-Glo^®^ analysis of cell viability under 0.25 and 1 μg/mL DOXO and DOXO-LIPO treatment for OVCAR-3 (**A**,**B**), Caov-3 (**C**,**D**), and SKOV3 (**E**,**F**) ovarian cancer cell lines. Data are mean + standard deviation (*n* = 3/5). Statistical analysis performed with two-way ANOVA. **** *p* < 0.0001, *** *p* < 0.001, ** *p* < 0.01, * *p* < 0.05. This section may be divided by subheadings. It should provide a concise and precise description of the experimental results, their interpretation, as well as the experimental conclusions that can be drawn.

**Table 1 cells-11-00824-t001:** Patient demographics for patient biopsy samples.

Pat. Code	Age	Stage Diagnosed	Diagnosis	Surgery	Ovarian Mass Location	Biopsy Location for AFM Analysis
ctrl 1	66	pelivic mass	benign fibroma	primary	left	right (contralateral normal ovary)
ctrl 2	77	pelivic mass	benign adenoma	primary	right	left (contralateral normal ovary)
ctrl 3	71	pelivic mass	benign fibroma	primary	left	right (contralateral normal ovary)
OC 1	71	HGSC IIIc	HGSC IIIc	interval	left/right	left
OC 2	69	HGSC IIIc	HGSC IIIc	interval	left/right	left
OC 3	59	HGSC IIIc	HGSC IIIc	interval	left/right	right

## Data Availability

Not applicable.

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
