# Peer review of "Mechanomimetic 3D Scaffolds as a Humanized In Vitro Model for Ovarian Cancer"

_cells, 2022, doi:10.3390/cells11050824_

Round 1
Reviewer 1 Report
The authors described the alteration of oncological behavior due to the alteration of the extracellular microenvironment. They specifically focused on ovarian cancer, featuring the mechanical properties of the normal human ovary and tissue stiffness of high-grade serous ovarian cancer (HGSC). They then mimicked these tissue properties by building collagen-based sponges. The fields of study are of particular interest to the scientific community.
However, this reviewer believes the work cannot be published in its current form.
Some points of attention:
The authors did not declare how many human tissue samples they analyzed to obtain the results shown in Figure 1. This is important because human samples usually show high variability.
Additionally, the authors must insert in fig 1 the histology of an adjacent slice tissue to the measured sample to show cellularity, morphology, and the presence of extracellular matrix.
How does the scaffold change after 7 and 10 days of cell culture? The authors show a change in stiffness, they should analyze the structure of the matrix to compare it with the structure of the scaffold before adding the cells. This analysis will allow us to identify the cell's active role in matrix remodeling.
In figure 3 a different ability of the cells to migrate and invade in the 3D matrix is evident. Authors should comment on this result.
Yap / taz RNA expression is not directly related to mechanosensor signaling. The authors should analyze the nuclear / cytoplasmic localization of these molecules in the different 3D samples.
Please in the paragraph "3D in vitro cytotoxic effect of free doxorubicin and doxorubicin-loaded liposomes" the authors should avoid listing all the changes for every single treatment but only highlight the differences in cellular responses inherent in the title free vs liposomes to meet the expectations of readers
Author Response
Reviewer 1:
The authors described the alteration of oncological behavior due to the alteration of the extracellular microenvironment. They specifically focused on ovarian cancer, featuring the mechanical properties of the normal human ovary and tissue stiffness of high-grade serous ovarian cancer (HGSC). They then mimicked these tissue properties by building collagen-based sponges. The fields of study are of particular interest to the scientific community.
However, this reviewer believes the work cannot be published in its current form.
Some points of attention:
- The authors did not declare how many human tissue samples they analyzed to obtain the results shown in Figure 1. This is important because human samples usually show high variability.
Additionally, the authors must insert in fig 1 the histology of an adjacent slice tissue to the measured sample to show cellularity, morphology, and the presence of extracellular matrix.
We thank the reviewer for bringing this to our attention. As per the reviewer’s suggestions, we updated the manuscript adding more information on the human samples tested. This catalysed a number of incremental changes to the manuscript as. Result of this comment, these are listed below.
- We have updated the Materials and Methods section with a ‘biopsy samples’ paragraph (page Page 5, line 226 onwards), as follows; .
Biopsy samples
Ethical approval for this study was obtained from NHS HRA Wales6 REC (15/WA/0065) to collect tissue samples from ovarian cancer patients and non-cancer controls. Formal written consent was obtained from all patients at the time of recruitment into the study. A total of 6 ovarian biopsies were collected for this study and processed for AFM analyses. The histological evaluation of the ovarian biopsies and the cancer diagnosis was confirmed by the Pathology Department as part of the patient’s routine clinical care. Three were used as normal ovary controls as they were obtained from normal contralateral ovaries of patients diagnosed with benign pathologies. The remaining 3 ovarian biopsy tissue samples represented HGSC stage IIIc.
Table 1. patient demographics for patient biopsy samples.
- We have updated the caption to Fig.1, to include the number of patients’ biopsies used. The caption (page 8; line 372) now reads as follows;
Young’s modulus (MPa) analysis of 3 HGCS III and 3 normal patient-derived biopsies by AFM.
- To further support the importance of the reviewer comment, we added Figure S1, highlighting interpatient variability as highlighted by AFM analysis:
FIGURE S1. AFM analysis of patients’ derived biopsies. (A) Young’s modulus (MPa) analysis of 3 HGCS III (OC1-2-3) and 3 normal patient-derived biopsies (ctrl 1-2-3) by AFM.
This Figure is referred to in the main manuscript body, in the Results section, paragraph 1 on page 7, line 349.
Patient biopsy mechanics, derived from both high-grade serous carcinoma (HGSC) stage III and normal ovary tissues, were analyzed at the nanoscale using AFM. A 5.5-fold increase in stiffness was observed between HGSC IIIa (0.11±0.034 MPa) and normal ovary (0.02±0.016 MPa) (p<0.05; Figure 1A, Figure S1).
- We have also added a second supplementary figure, Figure S2, reporting H&E staining for each patient used during the AFM analysis. This Figure is shown here for ease of reference.
FIGURE S1. H&E characterization of patients’ samples. (A H&E staining of 3 normal ovary biopsies samples. (B) H&E staining of 3 HGSC IIIc biopsies samples. Images were acquired with 4x and 20x objective.
This Figure is referred to in the main manuscript body, in the Results section, paragraph 1 on page 7, line 356.
H&E staining of patients’ biopsies samples was performed (Figure S2), reporting high presence of ECM/fibrotic tissue in the HGSC IIIc derived samples (Figure S2B).
In addition to the Figure and new data, we have updated the materials and methods section in order for the readers to understand the process and replicate in their own labs if desired. This section is included on page 5, line 241 onwards.
H&E patients’ biopsies staining
Paraffin sections were cut at 4 μm thickness. Hematoxylin and eosin staining was performed using the ST Infinity H&E Staining System (Leica Biosystems) in Leica Autostainer ST5010 XL. Paraffin was melted prior to staining by heating the slides at 60°C for 30 minutes, then slides were deparaffinized by performing 3 x 2-minute washes in xylene, 3 x 1-minute washes in 100% ethanol, and 1 x 1-minute wash in 95% ethanol before rinsing in tap water. Slides were incubated for 30 seconds in Hemalast, for 5 minutes in hematoxylin, and were rinsed for 1 minute in tap water. Next, slides were incubated for 30 seconds in Differentiator and 1 minute in Bluing agent, with each step followed by a tap water rinse for 1 minute then 95% ethanol for 1 minute. Slides were stained with eosin for 30 seconds, dehydrated in 95% ethanol for 1 minute, 4 minutes in 100% Ethanol, and 2 x 1 minute in 100% ethanol, and cleared for 3 x 2 minutes in xylene. Every step after the initial heating of the slides was done at room temperature.
- How does the scaffold change after 7 and 10 days of cell culture? The authors show a change in stiffness, they should analyze the structure of the matrix to compare it with the structure of the scaffold before adding the cells. This analysis will allow us to identify the cell's active role in matrix remodeling.
We thank the reviewer for this comment. Our analysisanalysis, however, aims to focus mainly on testing the suitability of the porous scaffold model, in replicating in vivo mechanical characteristics as well as evaluating cancer cell responses to matrix stiffness, highlighting proliferation and therapeutic sensitivity. It is not the aim of this manuscript to focus on OC cell matrix remodeling – we did not include other cell types associated with direct matrix remodeling. We are aware that solid tumors are typically stiffer than the surrounding tissue due to aberrant ECM deposition and organization that has a major influence on cell and tissue mechanics (Ingber DE. Mechanobiology and diseases of mechanotransduction. Ann Med. 2003; doi: 10.1080/07853890310016333; Pearce OMT et al. Deconstruction of a Metastatic Tumor Microenvironment Reveals a Common Matrix Response in Human Cancers. Cancer Discov. 2018 Mar; doi: 10.1158/2159-8290.)
In order to evaluate the ECM changes that can influence matrix softening, aside from cellular proliferation, as reported in the manuscript, we believe stromal cellular populaitons would need to be included in the study. In high grade serous ovarian cancers (HGSOC), representing the vast majority (75%) of total ovarian cancers, “Fibrosis” or “Mesenchymal” HGSOC molecular subtypes have been identified in all studies, systematically associated with poor patient survival, and characterized by high stromal content composed of myofbroblasts and ECM proteins, such as collagen and fibronectin. These factors are major causes of tumor stiffness (Mieulet, V. et al, Stiffness increases with myofibroblast content and collagen density in mesenchymal high grade serous ovarian cancer. Sci Rep 2021). It is known that tumor stiffness is associated with a high content in myofbroblasts, consistent with previous study in human HGSOC (Givel, A. M. et al. miR200-regulated CXCL12beta promotes fbroblast heterogeneity and immunosuppression in ovarian cancers. 2018, Nat. Commun.; Zhang, Q., et al. Cancer-associated stroma signifcantly contributes to the mesenchymal subtype signature of serous ovarian cancer. Gynecol. Oncol. 2019; Winterhof, B. J. et al. Single cell sequencing reveals heterogeneity within ovarian cancer epithelium and cancer associated stromal cells. Gynecol. Oncol. 2017; Yang, Z. et al. Reprogramming of stromal fbroblasts by SNAI2 contributes to tumor desmoplasia and ovarian cancer progression. Mol. Cancer, 2017). Myofbroblasts modulate tumor stiffness by secreting factors that bind to and remodel the ECM, such as matrix metalloproteinases and serpin proteins (Kharaishvili, G. et al. The role of cancer-associated fbroblasts, solid stress and other microenvironmental factors in tumor progression and therapy resistance. Cancer Cell Int., 2014; Tang, X. et al. Stromal miR-200s contribute to breast cancer cell invasion through CAF activation and ECM remodeling. Cell Death Difer., 2016). Importantly, it was also observed a correlation between high myofibroblast content and elevated collagen density, which correlates with tumor stiffness.
Fully characterizing matrix signature in this model would be interesting for future research purposes which could start from a proteomic characterization of patients’ biopsies matrisome and then evaluate if the changes in the 3D model will mirror some of those observations. Of course, to achieve a 3D in vitro model to investigate matrix changes during tumour progression, we envision the necessity of establishing a coculture of cancer cells and fibroblasts cells to include all the main players of ECM remodeling in a cancer tissue. (Naba A et al. Characterization of the Extracellular Matrix of Normal and Diseased Tissues Using Proteomics. J Proteome Res. 2017 doi: 10.1021/acs.jproteome.7b00191; Laklai, H. et al. Genotype tunes pancreatic ductal adenocarcinoma tissue tension to induce matricellular fibrosis and tumor progression. Nat Med (2016).).
This type of investigation would be very interesting and meaningful to pursue but it is beyond the scope of this research paper.
No change to the manuscript.
- In figure 3 a different ability of the cells to migrate and invade in the 3D matrix is evident. Authors should comment on this result.
We thank the reviewer for pointing this out. Following reviewer suggestions, we have updated the discussion (page 15, line 582) adding the following comment:
‘However, looking at the migration depth in the scaffolds recorded by immunofluores-cence staining, SKOV3 and Caov-3 showed higher invasiveness compared to OVCAR-3. Indeed, previous findings suggest that changes in stiffness of the cancer cell niche, as would be encountered by disseminated or metastatic OCCs, represents mechanism to further promote EMT [15]’
- Yap / taz RNA expression is not directly related to mechanosensor signaling. The authors should analyze the nuclear / cytoplasmic localization of these molecules in the different 3D samples.
We thank the reviewer for this comment. We are aware that YAP and its homolog transcriptional co-activator TAZ activate gene transcription through interaction with transcription factors in the nuclei and specifically, the ability of cells to perceive ECM mechanics and spread is associated to Hippo pathway effectors YAP and WWTR1 or TAZ shuttling to the nucleus, to exert their co-transcriptional activity.
While we acknowledge that the localization of YAP/TAZ would add value to the manuscript, doing this in 3D scaffold models of this type is challenging. In addition, our analysis focuses on the evaluation of the oncogenic potential of those two well-known mechanosensors in OC cells. It is important to point out that overexpression of YAP and TAZ is widespread in cancers such as pancreatic, colon rectal cancer, liver cancer and lung cancer, ovarian (Harvey KF, et al. The Hippo pathway and human cancer. Nat Rev Cancer. 2013; Johnson R, The two faces of Hippo: targeting the Hippo pathway for regenerative medicine and cancer treatment. Nat Rev Drug Discov. 2014, Yuan Y, et al. YAP overexpression promotes the epithelial-mesenchymal transition and chemoresistance in pancreatic cancer cells. Mol Med Rep 2016; Wang L, et al. Overexpression of YAP and TAZ is an independent predictor of prognosis in colorectal cancer and related to the proliferation and metastasis of colon cancer cells. PLoS One. 2013; Xu et al. Yes-associated protein is an independent prognostic marker in hepatocellular carcinoma. Cancer 2009; Wang et al., Overexpression of yes-associated protein contributes to progression and poor prognosis of non-small-cell lung cancer, 2010; Xia Y, et al. YAP promotes ovarian cancer cell tumorigenesis and is indicative of a poor prognosis for ovarian cancer patients. PLoS One. 2014).
These studies suggests that overexpressed YAP may interplay with b-catenin to drive proliferation of colon cancer cells, implying that YAP could play a role in cancer therapy (Rosenberg R, et al., Prognosis of patients with colorectal cancer is associated with lymph node ratio: a single-center analysis of 3,026 patients over a 25-year time period. Ann Surg, 2008; Zhang N et al. Merlin/NF2 tumor suppressor functions through the YAP oncoprotein to regulate tissue homeostasis in mammals. Dev Cell, 2010.). Overexpression of YAP was also found to enhance liver size and eventually lead to tumor development (Camargo FD, et al. YAP1 increases organ size and expands undifferentiated progenitor cells. Curr Biol 2017). YAP overexpression can induce anchorage-independent growth of non-transformed cell lines in soft agar, as well as growth-factor independent growth and epithelial to mesenchymal transition in MCF10A cells, YAP can also increase the ability of cultured epithelial cells to migrate (Overholtzer M, et al. Transforming properties of YAP, a candidate oncogene on the chromosome 11q22 amplicon. Proc Natl Acad Sci USA 2006). As such, YAP has been proposed as a candidate oncogene. Additionally, studies have shown that TAZ, and TAZ-dependent secretion of amphiregulin (AREG), also plays a significant role in breast tumorigenesis and metastasis: when overexpressed TAZ is knocked down in non-small-cell lung carcinoma (NSCLC), its proliferation and oncogenic properties are suppressed (Zhou Z, et al. TAZ is a novel oncogene in non-small cell lung cancer. Oncogene 2011).
All these studies evidence and demonstrate how YAP and TAZ display properties of classical oncogenes, suggesting that the assessment of their expression profile is informative. Indeed the RNA expression is helpful to understand if environmental changes such as stiffness can affect expression, as it has not been clearly demonstrated before and linked to ECM stifnes and/or mechanics.
As a result we have not endevoured to add the localization and protein staining to the current manuscript.
Is it established that SKOV3 cells showed the lowest expression of endogenous YAP and lowest activity while Caov3 and OVCAR3 reported a higher level of both YAP expression and its activity (Zhang, X., et al. The Hippo pathway transcriptional co-activator, YAP, is an ovarian cancer oncogene. Oncogene 2011). Interestingly our data reported an inversion of this trend on stiffer material, showing an increase of 2-fold in YAP expression on SKOV3 cultured in MS.
In line with reviewer comment, we updated the discussion section (page 15, line 606) to clarify the results of our analysis, as following:
‘Previous research reported that SKOV3 cells express a low amount of endogenous YAP (and lowest activity) compared to Caov3 and OVCAR3, which instead showed a higher level of both YAP expression and its activity [83]. Interestingly, we observed that while OVCAR-3 and Caov-3 downregulated YAP/TAZ expression on MS, SKOV3 cells in-creased the expression of both markers on MS compared to NS, suggesting a specific YAP/TAZ upregulation linked to high stiffness microenvironment.’
- Please in the paragraph "3D in vitro cytotoxic effect of free doxorubicin and doxorubicin-loaded liposomes" the authors should avoid listing all the changes for every single treatment but only highlight the differences in cellular responses inherent in the title free vs liposomes to meet the expectations of readers
We thank the reviewer for this comment. We modified and summarized the results (page 13, line 516), highlighting the main findings, as following :
‘
On 3D in vitro scaffolds, cells were initially treated with a dose range from 0.25 to 5 μg/ml DOXO and DOXO-LIPO (data not shown), and we chose to focus our analysis on a low and high dose close to the IC50 value (0.25 and 1 μg/ml) (Figure S5A-C). Overall, both 0.25 and 1 μg/ml of DOXO and DOXO-LIPO had a higher cytotoxic effect on 2D cultures rather than 3D MS and NS. DOXO-LIPO and DOXO show comparable effects among all cell lines tested, with the only exception of OVCAR-3 cells in 3D culture in which free DOXO had a greater cytotoxic effect compared to DOXO-LIPO (DOXO and DOXO-LIPO, 1 μg/ml, MS : 20% and 70% cell viability; NS: 46% and 98% cell viability). (Figure 6A-B). Interestingly, Caov-3 was more sensitive than the other two cell lines to both DOXO and DOXO-LIPO at 1 μg/ml, reporting a cytotoxic effect comparable to the one obtained in 2D (DOXO: 13%, 36%, and 30% and DOXO-LIPO: 29%, 34%, and 37% for 2D, MS, and NS, respectively) (Figure 6C-D). Finally, SKOV3 showed a DOXO and DOXO-LIPO, 1 μg/ml, MS : 33% and 50% cell viability; NS: 57% and 63% cell viability, respectively (Figure 6E,F). Empty liposomes had minimal effect on the proliferation of all cancer cell lines (Figure S4).
Reviewer 1:
The authors described the alteration of oncological behavior due to the alteration of the extracellular microenvironment. They specifically focused on ovarian cancer, featuring the mechanical properties of the normal human ovary and tissue stiffness of high-grade serous ovarian cancer (HGSC). They then mimicked these tissue properties by building collagen-based sponges. The fields of study are of particular interest to the scientific community.
However, this reviewer believes the work cannot be published in its current form.
Some points of attention:
- The authors did not declare how many human tissue samples they analyzed to obtain the results shown in Figure 1. This is important because human samples usually show high variability.
Additionally, the authors must insert in fig 1 the histology of an adjacent slice tissue to the measured sample to show cellularity, morphology, and the presence of extracellular matrix.
We thank the reviewer for bringing this to our attention. As per the reviewer’s suggestions, we updated the manuscript adding more information on the human samples tested. This catalysed a number of incremental changes to the manuscript as. Result of this comment, these are listed below.
- We have updated the Materials and Methods section with a ‘biopsy samples’ paragraph (page Page 5, line 226 onwards), as follows; .
Biopsy samples
Ethical approval for this study was obtained from NHS HRA Wales6 REC (15/WA/0065) to collect tissue samples from ovarian cancer patients and non-cancer controls. Formal written consent was obtained from all patients at the time of recruitment into the study. A total of 6 ovarian biopsies were collected for this study and processed for AFM analyses. The histological evaluation of the ovarian biopsies and the cancer diagnosis was confirmed by the Pathology Department as part of the patient’s routine clinical care. Three were used as normal ovary controls as they were obtained from normal contralateral ovaries of patients diagnosed with benign pathologies. The remaining 3 ovarian biopsy tissue samples represented HGSC stage IIIc.
Table 1. patient demographics for patient biopsy samples.
- We have updated the caption to Fig.1, to include the number of patients’ biopsies used. The caption (page 8; line 372) now reads as follows;
Young’s modulus (MPa) analysis of 3 HGCS III and 3 normal patient-derived biopsies by AFM.
- To further support the importance of the reviewer comment, we added Figure S1, highlighting interpatient variability as highlighted by AFM analysis:
FIGURE S1. AFM analysis of patients’ derived biopsies. (A) Young’s modulus (MPa) analysis of 3 HGCS III (OC1-2-3) and 3 normal patient-derived biopsies (ctrl 1-2-3) by AFM.
This Figure is referred to in the main manuscript body, in the Results section, paragraph 1 on page 7, line 349.
Patient biopsy mechanics, derived from both high-grade serous carcinoma (HGSC) stage III and normal ovary tissues, were analyzed at the nanoscale using AFM. A 5.5-fold increase in stiffness was observed between HGSC IIIa (0.11±0.034 MPa) and normal ovary (0.02±0.016 MPa) (p<0.05; Figure 1A, Figure S1).
- We have also added a second supplementary figure, Figure S2, reporting H&E staining for each patient used during the AFM analysis. This Figure is shown here for ease of reference.
FIGURE S1. H&E characterization of patients’ samples. (A H&E staining of 3 normal ovary biopsies samples. (B) H&E staining of 3 HGSC IIIc biopsies samples. Images were acquired with 4x and 20x objective.
This Figure is referred to in the main manuscript body, in the Results section, paragraph 1 on page 7, line 356.
H&E staining of patients’ biopsies samples was performed (Figure S2), reporting high presence of ECM/fibrotic tissue in the HGSC IIIc derived samples (Figure S2B).
In addition to the Figure and new data, we have updated the materials and methods section in order for the readers to understand the process and replicate in their own labs if desired. This section is included on page 5, line 241 onwards.
H&E patients’ biopsies staining
Paraffin sections were cut at 4 μm thickness. Hematoxylin and eosin staining was performed using the ST Infinity H&E Staining System (Leica Biosystems) in Leica Autostainer ST5010 XL. Paraffin was melted prior to staining by heating the slides at 60°C for 30 minutes, then slides were deparaffinized by performing 3 x 2-minute washes in xylene, 3 x 1-minute washes in 100% ethanol, and 1 x 1-minute wash in 95% ethanol before rinsing in tap water. Slides were incubated for 30 seconds in Hemalast, for 5 minutes in hematoxylin, and were rinsed for 1 minute in tap water. Next, slides were incubated for 30 seconds in Differentiator and 1 minute in Bluing agent, with each step followed by a tap water rinse for 1 minute then 95% ethanol for 1 minute. Slides were stained with eosin for 30 seconds, dehydrated in 95% ethanol for 1 minute, 4 minutes in 100% Ethanol, and 2 x 1 minute in 100% ethanol, and cleared for 3 x 2 minutes in xylene. Every step after the initial heating of the slides was done at room temperature.
- How does the scaffold change after 7 and 10 days of cell culture? The authors show a change in stiffness, they should analyze the structure of the matrix to compare it with the structure of the scaffold before adding the cells. This analysis will allow us to identify the cell's active role in matrix remodeling.
We thank the reviewer for this comment. Our analysisanalysis, however, aims to focus mainly on testing the suitability of the porous scaffold model, in replicating in vivo mechanical characteristics as well as evaluating cancer cell responses to matrix stiffness, highlighting proliferation and therapeutic sensitivity. It is not the aim of this manuscript to focus on OC cell matrix remodeling – we did not include other cell types associated with direct matrix remodeling. We are aware that solid tumors are typically stiffer than the surrounding tissue due to aberrant ECM deposition and organization that has a major influence on cell and tissue mechanics (Ingber DE. Mechanobiology and diseases of mechanotransduction. Ann Med. 2003; doi: 10.1080/07853890310016333; Pearce OMT et al. Deconstruction of a Metastatic Tumor Microenvironment Reveals a Common Matrix Response in Human Cancers. Cancer Discov. 2018 Mar; doi: 10.1158/2159-8290.)
In order to evaluate the ECM changes that can influence matrix softening, aside from cellular proliferation, as reported in the manuscript, we believe stromal cellular populaitons would need to be included in the study. In high grade serous ovarian cancers (HGSOC), representing the vast majority (75%) of total ovarian cancers, “Fibrosis” or “Mesenchymal” HGSOC molecular subtypes have been identified in all studies, systematically associated with poor patient survival, and characterized by high stromal content composed of myofbroblasts and ECM proteins, such as collagen and fibronectin. These factors are major causes of tumor stiffness (Mieulet, V. et al, Stiffness increases with myofibroblast content and collagen density in mesenchymal high grade serous ovarian cancer. Sci Rep 2021). It is known that tumor stiffness is associated with a high content in myofbroblasts, consistent with previous study in human HGSOC (Givel, A. M. et al. miR200-regulated CXCL12beta promotes fbroblast heterogeneity and immunosuppression in ovarian cancers. 2018, Nat. Commun.; Zhang, Q., et al. Cancer-associated stroma signifcantly contributes to the mesenchymal subtype signature of serous ovarian cancer. Gynecol. Oncol. 2019; Winterhof, B. J. et al. Single cell sequencing reveals heterogeneity within ovarian cancer epithelium and cancer associated stromal cells. Gynecol. Oncol. 2017; Yang, Z. et al. Reprogramming of stromal fbroblasts by SNAI2 contributes to tumor desmoplasia and ovarian cancer progression. Mol. Cancer, 2017). Myofbroblasts modulate tumor stiffness by secreting factors that bind to and remodel the ECM, such as matrix metalloproteinases and serpin proteins (Kharaishvili, G. et al. The role of cancer-associated fbroblasts, solid stress and other microenvironmental factors in tumor progression and therapy resistance. Cancer Cell Int., 2014; Tang, X. et al. Stromal miR-200s contribute to breast cancer cell invasion through CAF activation and ECM remodeling. Cell Death Difer., 2016). Importantly, it was also observed a correlation between high myofibroblast content and elevated collagen density, which correlates with tumor stiffness.
Fully characterizing matrix signature in this model would be interesting for future research purposes which could start from a proteomic characterization of patients’ biopsies matrisome and then evaluate if the changes in the 3D model will mirror some of those observations. Of course, to achieve a 3D in vitro model to investigate matrix changes during tumour progression, we envision the necessity of establishing a coculture of cancer cells and fibroblasts cells to include all the main players of ECM remodeling in a cancer tissue. (Naba A et al. Characterization of the Extracellular Matrix of Normal and Diseased Tissues Using Proteomics. J Proteome Res. 2017 doi: 10.1021/acs.jproteome.7b00191; Laklai, H. et al. Genotype tunes pancreatic ductal adenocarcinoma tissue tension to induce matricellular fibrosis and tumor progression. Nat Med (2016).).
This type of investigation would be very interesting and meaningful to pursue but it is beyond the scope of this research paper.
No change to the manuscript.
- In figure 3 a different ability of the cells to migrate and invade in the 3D matrix is evident. Authors should comment on this result.
We thank the reviewer for pointing this out. Following reviewer suggestions, we have updated the discussion (page 15, line 582) adding the following comment:
‘However, looking at the migration depth in the scaffolds recorded by immunofluores-cence staining, SKOV3 and Caov-3 showed higher invasiveness compared to OVCAR-3. Indeed, previous findings suggest that changes in stiffness of the cancer cell niche, as would be encountered by disseminated or metastatic OCCs, represents mechanism to further promote EMT [15]’
- Yap / taz RNA expression is not directly related to mechanosensor signaling. The authors should analyze the nuclear / cytoplasmic localization of these molecules in the different 3D samples.
We thank the reviewer for this comment. We are aware that YAP and its homolog transcriptional co-activator TAZ activate gene transcription through interaction with transcription factors in the nuclei and specifically, the ability of cells to perceive ECM mechanics and spread is associated to Hippo pathway effectors YAP and WWTR1 or TAZ shuttling to the nucleus, to exert their co-transcriptional activity.
While we acknowledge that the localization of YAP/TAZ would add value to the manuscript, doing this in 3D scaffold models of this type is challenging. In addition, our analysis focuses on the evaluation of the oncogenic potential of those two well-known mechanosensors in OC cells. It is important to point out that overexpression of YAP and TAZ is widespread in cancers such as pancreatic, colon rectal cancer, liver cancer and lung cancer, ovarian (Harvey KF, et al. The Hippo pathway and human cancer. Nat Rev Cancer. 2013; Johnson R, The two faces of Hippo: targeting the Hippo pathway for regenerative medicine and cancer treatment. Nat Rev Drug Discov. 2014, Yuan Y, et al. YAP overexpression promotes the epithelial-mesenchymal transition and chemoresistance in pancreatic cancer cells. Mol Med Rep 2016; Wang L, et al. Overexpression of YAP and TAZ is an independent predictor of prognosis in colorectal cancer and related to the proliferation and metastasis of colon cancer cells. PLoS One. 2013; Xu et al. Yes-associated protein is an independent prognostic marker in hepatocellular carcinoma. Cancer 2009; Wang et al., Overexpression of yes-associated protein contributes to progression and poor prognosis of non-small-cell lung cancer, 2010; Xia Y, et al. YAP promotes ovarian cancer cell tumorigenesis and is indicative of a poor prognosis for ovarian cancer patients. PLoS One. 2014).
These studies suggests that overexpressed YAP may interplay with b-catenin to drive proliferation of colon cancer cells, implying that YAP could play a role in cancer therapy (Rosenberg R, et al., Prognosis of patients with colorectal cancer is associated with lymph node ratio: a single-center analysis of 3,026 patients over a 25-year time period. Ann Surg, 2008; Zhang N et al. Merlin/NF2 tumor suppressor functions through the YAP oncoprotein to regulate tissue homeostasis in mammals. Dev Cell, 2010.). Overexpression of YAP was also found to enhance liver size and eventually lead to tumor development (Camargo FD, et al. YAP1 increases organ size and expands undifferentiated progenitor cells. Curr Biol 2017). YAP overexpression can induce anchorage-independent growth of non-transformed cell lines in soft agar, as well as growth-factor independent growth and epithelial to mesenchymal transition in MCF10A cells, YAP can also increase the ability of cultured epithelial cells to migrate (Overholtzer M, et al. Transforming properties of YAP, a candidate oncogene on the chromosome 11q22 amplicon. Proc Natl Acad Sci USA 2006). As such, YAP has been proposed as a candidate oncogene. Additionally, studies have shown that TAZ, and TAZ-dependent secretion of amphiregulin (AREG), also plays a significant role in breast tumorigenesis and metastasis: when overexpressed TAZ is knocked down in non-small-cell lung carcinoma (NSCLC), its proliferation and oncogenic properties are suppressed (Zhou Z, et al. TAZ is a novel oncogene in non-small cell lung cancer. Oncogene 2011).
All these studies evidence and demonstrate how YAP and TAZ display properties of classical oncogenes, suggesting that the assessment of their expression profile is informative. Indeed the RNA expression is helpful to understand if environmental changes such as stiffness can affect expression, as it has not been clearly demonstrated before and linked to ECM stifnes and/or mechanics.
As a result we have not endevoured to add the localization and protein staining to the current manuscript.
Is it established that SKOV3 cells showed the lowest expression of endogenous YAP and lowest activity while Caov3 and OVCAR3 reported a higher level of both YAP expression and its activity (Zhang, X., et al. The Hippo pathway transcriptional co-activator, YAP, is an ovarian cancer oncogene. Oncogene 2011). Interestingly our data reported an inversion of this trend on stiffer material, showing an increase of 2-fold in YAP expression on SKOV3 cultured in MS.
In line with reviewer comment, we updated the discussion section (page 15, line 606) to clarify the results of our analysis, as following:
‘Previous research reported that SKOV3 cells express a low amount of endogenous YAP (and lowest activity) compared to Caov3 and OVCAR3, which instead showed a higher level of both YAP expression and its activity [83]. Interestingly, we observed that while OVCAR-3 and Caov-3 downregulated YAP/TAZ expression on MS, SKOV3 cells in-creased the expression of both markers on MS compared to NS, suggesting a specific YAP/TAZ upregulation linked to high stiffness microenvironment.’
- Please in the paragraph "3D in vitro cytotoxic effect of free doxorubicin and doxorubicin-loaded liposomes" the authors should avoid listing all the changes for every single treatment but only highlight the differences in cellular responses inherent in the title free vs liposomes to meet the expectations of readers
We thank the reviewer for this comment. We modified and summarized the results (page 13, line 516), highlighting the main findings, as following :
‘
On 3D in vitro scaffolds, cells were initially treated with a dose range from 0.25 to 5 μg/ml DOXO and DOXO-LIPO (data not shown), and we chose to focus our analysis on a low and high dose close to the IC50 value (0.25 and 1 μg/ml) (Figure S5A-C). Overall, both 0.25 and 1 μg/ml of DOXO and DOXO-LIPO had a higher cytotoxic effect on 2D cultures rather than 3D MS and NS. DOXO-LIPO and DOXO show comparable effects among all cell lines tested, with the only exception of OVCAR-3 cells in 3D culture in which free DOXO had a greater cytotoxic effect compared to DOXO-LIPO (DOXO and DOXO-LIPO, 1 μg/ml, MS : 20% and 70% cell viability; NS: 46% and 98% cell viability). (Figure 6A-B). Interestingly, Caov-3 was more sensitive than the other two cell lines to both DOXO and DOXO-LIPO at 1 μg/ml, reporting a cytotoxic effect comparable to the one obtained in 2D (DOXO: 13%, 36%, and 30% and DOXO-LIPO: 29%, 34%, and 37% for 2D, MS, and NS, respectively) (Figure 6C-D). Finally, SKOV3 showed a DOXO and DOXO-LIPO, 1 μg/ml, MS : 33% and 50% cell viability; NS: 57% and 63% cell viability, respectively (Figure 6E,F). Empty liposomes had minimal effect on the proliferation of all cancer cell lines (Figure S4).
Reviewer 2 Report
This paper is interesting and valuable for researchers on biomaterials, scaffolds, tissue engineering, cancer biology, or pharmaceutics. However, the authors should clarify the importance of scaffold in terms of the environmental difference between in vitro and in vivo. The readers must be confusing and feel difficult to follow. Taken together, major revisions should be made before re-submission. The paper would be re-considered only when all the comments were responded.
- Introduction
The interaction between cells and scaffold composed of biomaterial should be clarified. Scaffolds can assist the ECM composition, leading to the enhancement of cell function. The results are interesting, so the authors should add the paragraph on various tissue regions for readers’ better understanding. To reduce the authors’ burden, I suggest at least these recent papers be added for revision.
[1] For overall concept Int. J. Mol. Sci. 2021, 22(16), 8657
[2] For bone tissue engineering RSC Adv., 2019, 9, 26252
[3] For skin tissue engineering Advanced Drug Delivery Reviews 128 (2011) 352–366
[4] For cancer tissue engineering Cancers 2020, 12(10), 2754
[5] For cardiac tissue engineering Macromol. Biosci. 2018, 18, 1800079
[6] For muscle tissue engineering Current Opinion in Biotechnology 2017, 47:16–22
2.
Are these in vitro results linked, or are they similar to in vivo? Can the authors prove? The points are so important.
3.
How about the degradation profile of the scaffold?
Author Response
Reviewer 2:
This paper is interesting and valuable for researchers on biomaterials, scaffolds, tissue engineering, cancer biology, or pharmaceutics. However, the authors should clarify the importance of scaffold in terms of the environmental difference between in vitro and in vivo. The readers must be confusing and feel difficult to follow. Taken together, major revisions should be made before re-submission. The paper would be re-considered only when all the comments were responded.
- Introduction
The interaction between cells and scaffold composed of biomaterial should be clarified. Scaffolds can assist the ECM composition, leading to the enhancement of cell function. The results are interesting, so the authors should add the paragraph on various tissue regions for readers’ better understanding. To reduce the authors’ burden, I suggest at least these recent papers be added for revision.
[1] For overall concept Int. J. Mol. Sci. 2021, 22(16), 8657
[2] For bone tissue engineering RSC Adv., 2019, 9, 26252
[3] For skin tissue engineering Advanced Drug Delivery Reviews 128 (2011) 352–366
[4] For cancer tissue engineering Cancers 2020, 12(10), 2754
[5] For cardiac tissue engineering Macromol. Biosci. 2018, 18, 1800079
[6] For muscle tissue engineering Current Opinion in Biotechnology 2017, 47:16–22
We thank the reviewer for giving us the chance to expand the background on this important topic. As a result and as suggested, we have updated the introduction section (page 3, from line 98 133), as follows;
Tissue engineering (TE) describes the process of fabricating functional 3D tissues using a combination of scaffolds and/or devices with cells, to facilitate essential cellular functions such as growth, differentiation, migration, and organization [49]. In the field of regenerative medicine, those 3D devices have the aim to replace or "regenerate" human cells, tissues or organs to restore or establish normal function [50]. As established by decades of research on tissue engineering (TE) manufacturing, in order to create an effective 3D construct three crucial components (called TE triad) are needed: a relevant selection of cells, a biomaterial scaffold which provide the structural support for cell attachment and guide tissue development [51, 52], and chemicals and biophysical signals that crosstalk to ultimately recreate tissue [53, 54]. Typically, in tissue engineering, three individual groups of biomaterials are used in the fabrication of scaffolds: ceramics, synthetic polymers and natural polymers [49], and they have been explored for a variety of applications, tissue engineering bone [55], skin [56], cardiac tissue [57], skeletal muscle [58] and cancer models [59] . Natural biomaterials are bioactive, biodegradable and allow host cells to produce their own extracellular matrix and remodel the scaffold [49]. However, depending on the scaffold employed they generally have poor mechanical properties, which limits their use in, for example, load bearing orthopedic applications.
For its high biocompatibility and bioactivity, natural polymer collagen was selected for scaffolds fabrication in this research. Collagen is the most abundant structural protein in the connective tissues and its homology across species provides low antigenicity and high biocompatibility [60, 61]; in humans, collagen represents one-third of the total protein content in the body [62]. Over the last two decades, four major scaffolding approaches for TE have evolved : pre-made porous scaffolds, decellularized extracellular matrix (ECM), cell sheets with self-secreted ECM, Cell encapsulation in self-assembled hydrogel matrix [63]. Among those, the most common approach is the use of a pre-made porous scaffold [54], since it harbours a number of advantages: it has the most diversified range of biomaterials available to use -natural or synthetic-[64]; precise architectural features and microstructures can be incorporated [65]; physicochemical characteristics can be tuned to mimic the physical properties of native tissues [66]. Specifically, matrix stiffness cues can be easily tuned in porous collagen type I-based interconnected scaffold systems, by varying crosslinking types or percentages [67-70], to control porosity and fiber organization resulting in a tunable system for 3D mechanical studies [71, 72]. Indeed, easy to reproduce, convenient to handle, and amenable to large-scale use, porous scaffolds now have a wide scope of applications [73-76]. However, only a few solid tumours have been tested using these approaches, i.e. breast, prostate, glioblastoma mainly investigated using chitosan-alginate or chitosan-hyaluronic acid based-scaffolds [77-83].
- Are these in vitro results linked, or are they similar to in vivo? Can the authors prove? The points are so important.
We thank the reviewer for giving us the chance to discuss deeper this topic referring to literature. Indeed, we believe that these in vitro results are similar to studies that have revealed interactions between mechanics, proliferation and stiffness in vivo. There is a large body of evidence to validate this statement and selected examples have been added or previously woven into the discussion at certain sections to improve its relevance and the wider engagement of the audience.
MECHANIC:
As first link with in vivo evidences, the AFM range established for the scaffolds under investigations derives from analysis performed on native tissue derived from biopsies from patients with High-Grade Serous Carcinoma stage III and normal ovary.
Previous research analyzed tumor stiffness in vivo, measured by an innovative technology called Supersonic shear wave elastography, in a Patient-Derived Xenografts (PDX) mouse models engrafted with HGSOC tumors isolated from patients. The results showed that tumor stiffness significantly increased and reached 120 to 140 kPa over time in Mesenchymal HGSOC, while it remained low (not higher than 60 kPa) in Non-Mesenchymal tumors. In high grade serous ovarian cancers (HGSOC), representing the vast majority (75%) of total ovarian cancers, “Fibrosis” or “Mesenchymal” HGSOC molecular subtype has been identified in all studies and is systematically associated with poor patient survival, and characterized by high stromal content composed of myofbroblasts and ECM proteins, such as collagen and fibronectin, major causes of tumor stiffness (Mieulet, V. et al, Stiffness increases with myofibroblast content and collagen density in mesenchymal high grade serous ovarian cancer. Sci Rep 2021). Furthermore, in few cases, they observed a new tumor nodule emerging from a stiff Mesenchymal tumor, interestingly, the new nodule—of little size—was softer than the established initial tumor, suggesting tumor proliferation starts with an initial soft state.
Besides, our data are in line with observations made in other cancers.
In breast cancer, malignant tissue is typically stiffer than its normal counterpart with studies showing that normal breast tissue is 20 times softer than its neoplastic counterpart (Seewaldt, V., ECM stiffness paves the way for tumor cells. Nature Medicine, 2014. 20(4): p. 332-333).
Another example is the elastic moduli of healthy thyroid tissue (9.0–11.4 kPa) which can increase by a full order of magnitude to 44–110 kPa in patients with papillary adenocarcinoma (Lyshchik, A. et al. Elastic moduli of thyroid tissues under compression. Ultrason. Imaging, 2005; Guimarães, C.F., Gasperini, L., Marques, A.P. et al. The stiffness of living tissues and its implications for tissue engineering. Nat Rev Mater).
Regarding rheological analysis, storage modulus of MS and NS scaffolds spans in range of stiffness reported in the literature with reports on tissues/organs stiffness ranging from 0.2-64 kPa (Guimarães, C.F., Gasperini, L., Marques, A.P. et al. The stiffness of living tissues and its implications for tissue engineering. Nat Rev Mater, 2020; Park JS et al., The effect of matrix stiffness on the differentiation of mesenchymal stem cells in response to TGF. Biomaterials., 2011; Asano S, et al., Matrix stiffness regulates migration of human lung fibroblasts. Physiol Rep. 2017).
Other examples derive from studies on lung fibrosis were the elastic modulus of the fibrotic lung (15–100 kPa) is much stiffer than that of normal lung parenchyma (0.5–5 kPa) (Liu, F., J. D. Mih, B. S. Shea, A. T. Kho, A. S. Sharif, A. M.Tager, et al. 2010. Feedback amplification of fibrosis through matrix stiffening and COX-2 suppression. J. CellBiol. 190:693–706.; Booth, A. J., R. Hadley, A. M. Cornett, A. A. Dreffs, S. A.Matthes, J. L. Tsui, et al. 2012. Acellular normal and fibrotic human lung matrices as a culture system for in vitro investigation. Am. J. Respir. Crit. Care Med. 186:866–876).
We modified and summarized the discussion above at page 15, line 566-585 as following:
‘Our results are in line with previous research which analyzed tumor stiffness in vivo. For example, researchers employed Supersonic shear wave elastography in a Patient-Derived Xenografts (PDX) mouse models engrafted with HGSOC tumors isolated from patients, recording a significantly increase in tumor stiffness (120 to 140 kPa) over time in Mesenchymal HGSOC, while it remained low (not higher than 60 kPa) in Non-Mesenchymal tumors. In high grade serous ovarian cancers (HGSOC), representing the vast majority (75%) of total ovarian cancers, “Fibrosis” or “Mesenchymal” HGSOC molecular subtype has been identified in all studies and is systematically associated with poor patient survival. It is characterized by high stromal content composed of myofibroblasts and ECM proteins, such as collagen and fibronectin, major causes of tumor stiffness [102]. In the same study, in few cases, they observed a new tumor nodule emerging from a stiff Mesenchymal tumor, interestingly, the new nodule—of little size—was softer than the established initial tumor, suggesting tumor proliferation could origin with an initial soft state. Besides, our data are in line with observations made in other cancer types. In breast cancer, malignant tissue is typically stiffer than its normal counterpart with studies showing that normal breast tissue is 20 times softer than its neoplastic counterpart [103]; elastic moduli of healthy thyroid tissue (9.0–11.4 kPa) can increase by a full order of magnitude to 44–110 kPa in patients with papillary adenocarcinoma [104]. Furthermore, storage modulus of MS and NS scaffolds spans in range of stiffness reported in the literature referring tissues/organs stiffness ranging from 0.2-64 kPa [105-107].’
PROLIFERATION & STIFFNESS:
In line with our results, previous studies found that increasing substrate stiffness promoted the proliferation of SKOV-3 cells (Fan Y, et al. Substrate Stiffness Modulates the Growth, Phenotype, and Chemoresistance of Ovarian Cancer Cells. Front Cell Dev Biol. 2021).
We recorded that an increase in proliferation on MS scaffolds for SKOV3 cells and OVCAR-3 led to softening of the scaffold after 7 days of culture. This could be linked to two phenomena reported in literature. First, cancer cells are physically softer than normal cells (Lekka, M. Discrimination between normal and cancerous cells using AFM. 2016 Bionanoscience; Alibert, C. et al. Are cancer cells really softer than normal cells? Biol. Cell 2017) and metastatic cancer cells are more mechanically compliant than their non-metastatic counterparts (Li, et al.AFM indentation study of breast cancer cells. Biochem. Biophys. Res. Commun. 2008; Park, S. Nano-mechanical phenotype as a promising biomarker to evaluate cancer development, progression, and anti-cancer drug efficacy. J. Cancer Prev 2016); contributing to the overall softening of the tissue.
Second, previous findings suggest that a reduction in the stiffness of the cancer cell niche, as would be encountered by disseminated or metastatic OCCs, is a mechanism to promote EMT (Fan Y, et al. Substrate Stiffness Modulates the Growth, Phenotype, and Chemoresistance of Ovarian Cancer Cells. Front Cell Dev Biol. 2021), suggesting that the progressive softening of the matrix is a crucial step to promote metastasis.
To better discuss this point, we edited the discussion section (page 15-16 lines 600-619) as following:
‘For this reason, we selected relevant HGSC and non-serous tumor derived ovarian cancer cell lines to investigate mechanosensing behavior in response to different 3D biomechanic scaffolds. All cell lines employed colonized and were viable on both MS and NS with no changes in morphology. However, looking at the migration depth in the scaffolds recorded by immunofluorescence staining, SKOV3 and Caov-3 showed higher invasiveness compared to OVCAR-3. Indeed, previous findings suggest that changes in stiffness of the cancer cell niche, as would be encountered by disseminated or metastatic OCCs, represents mechanism to further promote EMT [15]. A significant difference was recorded in the proliferation rate, showing higher proliferation of OVCAR-3 on NS while both Caov-3 and SKOV3 had higher proliferation rates on MS, suggesting higher responsiveness to rigid substrates. The higher rate of proliferation of OVCAR-3 and SKOV3 compared to Caov-3 resulted in a lower storage modulus after 7 days of culturing for MS, suggesting a link between cells proliferation and softening of the scaffolds recorded with rheometry. This phenomenon could be linked to two phenomena reported in literature. First, cancer cells are physically softer than normal cells [112, 113] and metastatic cancer cells are more mechanically compliant than their non-metastatic counterparts [114, 115]; contributing to the overall softening of the tissue. Second, pre-vious findings suggest that a reduction in the stiffness of the cancer cell niche, as would be encountered by disseminated or metastatic OCCs, is a mechanism to promote EMT [116], suggesting that the progressive softening of the matrix is a crucial step to promote metastasis.’
TREATMENT SENSITIVITY:
Other reports in literature showed that ovarian cancer cells grown on softer substrates with a lower elastic modulus were less sensitive to chemotherapeutic agents (Fan Y, et al. Substrate Stiffness Modulates the Growth, Phenotype, and Chemoresistance of Ovarian Cancer Cells. Front Cell Dev Biol. 2021; McGrail DJ, et al. The malignancy of metastatic ovarian cancer cells is increased on soft matrices through a mechanosensitive Rho-ROCK pathway. J Cell Sci. 2014). In this research, mRNA microarray analysis showed that platinum drug-resistance genes including ERBB2, BCL-2, MAP3K5, PIK3R1, and BIRC3 are significantly upregulated in SKOV-3 cells on soft substrates. Furthermore, on softer gels cells expressed more ABCB1 and ABCB4, ABC transporter and the first to be identified to selectively confer MDR by directly pumping out anticancer drugs, including paclitaxel, doxorubicin, topotecan, docetaxel (Juliano and Ling, 1976; Bourhis et al., 1989; Veneroni et al., 1994). These results suggest that EMT promotes chemoresistance in SKOV-3 cells on soft substrates via the upregulation of ABCB1 and ABCB4 (Fan Y, et al. Substrate Stiffness Modulates the Growth, Phenotype, and Chemoresistance of Ovarian Cancer Cells. Front Cell Dev Biol. 2021)
Similar results were observed in other tumors like breast cancer, where stiff ECM enhanced proapoptotic JNK activity to sensitize cells to treatment, whereas a soft ECM promoted treatment resistance by elevating NF-κB activity and compromising JNK activity (Drain AP, et al. Matrix compliance permits NF-κB activation to drive therapy resistance in breast cancer. J Exp Med. 2021); while in other tumour like pancreatic cancer stiffness induces chemoresistance to paclitaxel, but not to gemcitabine, both commonly used therapeutics (Rice, A. et al. Matrix stiffness induces epithelial–mesenchymal transition and promotes chemoresistance in pancreatic cancer cells. Oncogenesis, 2017).
We think this point was discussed at page 16 lines 641-648. No changes in the manuscript were made referred to this.
- How about the degradation profile of the scaffold?
We thank the reviewer for this comment.
We are aware that collagen-based scaffolds hold great potential for tissue engineering, for both bone and soft tissue engineering to replace damaged or lost tissues, since these biomaterials provide an environment close to the native extracellular matrix (O’Brien, F.J. Biomaterials & scaffolds for tissue engineering. Mater. Today 2011; Christman, K.L. Biomaterials for tissue repair. Science 2019). For this purpose collagen-based scaffolds should at least provide: (1) high biocompatibility, (2) a highly porous structure with interconnected pores to allow influx of progenitor cells and blood vessels, (3) mechanical properties similar to the native tissue and (4) degradation properties and kinetics that match the characteristic speed of the tissue to be regenerated (Caballé-Serrano, J.; et al. Tissue Integration and Degradation of a Porous Collagen-Based Scaffold Used for Soft Tissue Augmentation. Materials 2020)
Since our application doesn’t envision any implant or regenerative medicine applications, we decided to not include degradation studies in our manuscript. We performed a test with collagenase type I, evaluating the weight loss at 2-24-48h. NS scaffolds were completely digested by the enzyme after 2h while MS scaffolds showed a weight loss of around 20-28% at 24 and 48h.
Furthermore, we extensively tested porous collagen-based scaffolds in many previous publications for bone and cartilage regeneration studies; the scaffold production protocol it’s also a patented protocol currently used in GMP facility for translational studies (Bauza G et al., Improving the immunosuppressive potential of articular chondroprogenitors in a three-dimensional culture setting. Sci Rep., 2020; Silvia Minardi et al. Evaluation of the osteoinductive potential of a bio-inspired scaffold mimicking the osteogenic niche for bone augmentation, Biomaterials,2015; Corradetti B et al, Chondroitin Sulfate Immobilized on a Biomimetic Scaffold Modulates Inflammation While Driving Chondrogenesis. Stem Cells Transl Med. 2016).
No changes in the manuscript.

Round 2
Reviewer 1 Report
In the revised version the Authors commented on all the points of the reviewer. However, HE images are neither included in Figure 1 of the minimum text nor in the supplementary materials. They have to include them before acceptance
Author Response
The authors thank the reviewer for the help and precious suggestions that improved the quality of our manuscript.
Reviewer 2 Report
The authors have responded to all the comments.
I recommend the publication.
Author Response
In the revised version the Authors commented on the reviewer 1 comment: However, HE images are neither included in Figure 1 of the minimum text nor in the supplementary materials. They have to include them before acceptance. We thank the reviewer for pointing this out. As requested, we updated the manuscript in the results and supplementary figures section. A section referring to the H&E in the results was added at page 7 p. 357-359, as following: ‘H&E staining of patients’ biopsies samples was performed (Figure S2), reporting high presence of ECM/fibrotic tissue in the HGSC IIIc derived samples (Figure S2B).’ We added H&E analysis as supplementary figure Figure S2, reporting H&E staining for each patient used during the AFM analysis. This Figure is shown here for ease of reference.
